# Isotropy in the Contextual Embedding Space: Clusters and Manifolds

**Xingyu Cai, Jiaji Huang, Yuchen Bian, Kenneth Church**
Baidu Research, 1195 Bordeaux Dr, Sunnyvale, CA 94089, USA
{xingyucai,huangjiaji,yuchenbian,kennethchurch}@baidu.com

## Abstract

The geometric properties of contextual embedding spaces for deep language models such as BERT and ERNIE, have attracted considerable attention in recent years. Investigations on the contextual embeddings demonstrate a strong anisotropic space such that most of the vectors fall within a narrow cone, leading to high cosine similarities. It is surprising that these LMs are as successful as they are, given that most of their embedding vectors are as similar to one another as they are. In this paper, we argue that the isotropy indeed exists in the space, from a different but more constructive perspective. We identify isolated clusters and low dimensional manifolds in the contextual embedding space, and introduce tools to both qualitatively and quantitatively analyze them. We hope the study in this paper could provide insights towards a better understanding of the deep language models.

## 1 Introduction

The polysemous English word "bank" has two common senses: 1. the money sense, a place that people save or borrow money; 2. the river sense, a slope of earth that prevents the flooding. In modern usage, the two senses are very different from one another, though interestingly, both senses share similar etymologies (and both can be traced back to the same word in Proto-Germanic). In the **static embedding**, multiple instances of the same word (e.g. "bank") will be represented using the same vector. On the contrary, the **contextual embedding** assigns different vectors to different instances of the same word, depending on the context. Historically, static embedding models like Word2vec (Mikolov et al., 2013b) and GloVe (Pennington et al., 2014), predated contextual embedding models such as ELMo (Peters et al., 2018), GPT (Radford et al., 2018), BERT (Devlin et al., 2018) and ERNIE (Sun et al., 2019). Much of the literature on language modeling has moved to contextual embeddings recently, largely because of their superior performance on the downstreaming tasks.

### 1.1 Related Work

The static embeddings are often found to be easier to interpret. For example, the Word2Vec and GloVe papers discuss adding and subtracting vectors, such as: vec(king) - vec(man) + vec(women) = vec(queen). Inspired by this relationship, researchers started to explore geometric properties of static embedding spaces. For example, Mu & Viswanath (2018) proposed a very counter-intuitive method that removes the top principle components (the dominating directions in the transformed embedding space), which surprisingly improved the word representations. Rather than completely discarding the principle components, Liu et al. (2019) proposed to use a technique called Conceptor Negation, to softly suppress transformed dimensions with larger variances. Both approaches, simply removing certain principle components as well as Conceptor Negation, produce significant improvements over vanilla embeddings obtained by static language models. In Huang et al. (2020), the authors studied how to effectively transform static word embeddings from one language to another.

Unfortunately, the strong illustrative representation like the king-queen example above, is no longer obvious in a general contextual embedding space. Arguing that syntax structure indeed exists in the contextual embeddings, Hewitt & Manning (2019) proposed a structural probe to identify the syntax trees buried in the space, and found the evidence of implicit syntax tree in BERT and ELMo. The advantage of contextual embedding over the static counterpart, mainly come from its capability to assign different vectors to the same word, depending on the word sense in the context. Researchers in (Reif et al., 2019) found such a geometric representation of word senses in the BERT model. These papers reveal the existence of linguistic features embedded implicitly in the contextual vector spaces.

The geometric properties of contextual embedding space are also investigated and compared with the static embedding space. Mimno & Thompson (2017) found anisotropy when negative sampling is used. In (Ethayarajh, 2019), the authors characterize how vectors are distributed in the contextual space. They found that most vectors occupy in a relatively narrow cone in the space. Pairs of vectors within this cone have large cosines. This phenomenon can be found in most state-of-the-art contextual embedding models. In (Gao et al., 2019), the authors named this phenomenon "representation degeneration", and attempted to mitigate the problem by introducing a regularization term that minimizes cosine similarities between vectors. In a very recent work, Demeter et al. (2020) suggest there is a structure weakness in the space that leads to bias when using soft-max, as is common with deep language models.

## 1.2 MOTIVATION AND CONTRIBUTIONS

Isotropy often makes the space more effectively utilized and more robust to perturbations (no extreme directions that lead to high condition number). It is counter-intuitive and not clear why those contextual embedding models perform remarkably well on many tasks given their anisotropic embeddings bring all the vectors close together, hard to distinguish one from another. On one hand, it is widely believed that contextual embeddings encode the relevant linguistic information (e.g. (Reif et al., 2019)), but on the other hand, it is also widely believed that the contextual space is anisotropic that representations become degenerated (e.g. (Mimno & Thompson, 2017), (Gao et al., 2019), (Ethayarajh, 2019)). These motivate us to find a reasonable understanding that bridges this gap.

This paper is similar in spirit to (Mu & Viswanath, 2018), but different in three aspects. First, we generalize their work on traditional static embeddings to more modern contextual embeddings. Second, we introduce clustering methods to isolate the space, whereas they used PCA to remove dominant dimensions (that tend to dominate the variance). Finally, we identify low dimensional manifolds in the space, and introduce an alternative approach (LID) to characterize local subspaces.

**Key Contributions**: This paper takes a deeper look into the contextual embedding spaces of popular pre-trained models. It identifies the following facts that were misunderstood or not known before: 1) We find isotropy within clusters in the contextual embedding space, in contrast to previous reports of anisotropy (caused by misleading isolated clusters). We introduce clustering and center shifting to reveal the isotropy, and show more consistent layer-wise behavior across models. 2) We find a Swiss-Roll manifold in GPT/GPT2 embeddings, but not in BERT/DistilBERT embeddings. The manifold is related to word frequency, suggesting a difference in how models evolve as they see more data. We use approximate Local Intrinsic Dimension (LID) to characterize the manifold, and find contextual embedding models, including all BERT, GPT families and ELMo, often have small LIDs. The small LIDs can be viewed as the local anisotropy of the space. The code for this paper could be found at `https://github.com/TideDancer/IsotropyContxt`.

## 2 ANALYSIS SETTINGS

### 2.1 MODELS AND DATASETS

In this paper, we consider popular pre-trained contextual embedding models, including BERT, DistilBERT (Sanh et al., 2019) (or denoted as D-BERT in the rest of the paper), GPT, GPT2 (Radford et al., 2019) and ELMo. For the BERT and GPT families, we perform our evaluations on the pre-trained uncased base models from Huggingface (https://huggingface.co/transformers/index.html#). The pre-trained ELMo model is from AllenNLP (https://docs.allennlp.org/v1.0.0/). BERT and D-BERT are **non-causal** models because of their attention mechanism, where tokens can attend to any token in the input, regardless of their relative positions. In contrast, GPT and GPT2 are **causal** models because attention is limited to the tokens previously seen in the input.

Different models achieve contextual embedding in different ways. For instance, BERT adds positional embeddings to the token embeddings, while ELMo performs vector concatenation. Most models start with an initial layer that maps token ids to vectors. This paper is not concerned with that lookup table layer, and only focuses on the layers after that. The base BERT, GPT and GPT2 models have 12 layers of interest, indexed from 0 to 11, while D-BERT has 6 layers and ELMo has two.

We use Penn Tree Bank (PTB) (Marcus et al., 1993) and WikiText-2 (Merity et al., 2016) datasets. The PTB has 0.88 million words and WikiText-2 has 2 million. Both of them are the standard datasets

for language models. In the rest of the paper, we report on PTB since we see similar results with both datasets. Details on WikiText-2 analysis could be found in Appendix.

## 2.2 NOTATION

For each position in a corpus, we have a **word**. Words are converted into **tokens**, using the appropriate tokenizer for the model. Tokenizers could split some words into subwords, therefore, the number of obtained tokens (denoted as $n$) could be more than number of words in the corpus. PTB, for example, contains 0.88 million words, but has $n = 1.2$ million tokens, when processed by BERT's tokenizer. Let $V$ be the **vocabulary**, a set of distinct tokens. For any element in the vocabulary $V$, we call it a **type**. For example, BERT has a vocabulary of roughly 30,000 types. We may mix using "word" and "type" for ease of reading. We denote the $i$-th type in $V$ as $t_i$. Let $\Phi(t_i) = \{\phi_1(t_i), \phi_2(t_i), \ldots\}$ be the set of all embedding instances of $t_i$ (note that different contexts in the corpus yield different embeddings of $t_i$). By construction, $\sum_t |\Phi(t)| = n$. We define the **inter-type** cosine similarity as

$$S_{\text{inter}} \triangleq \mathbb{E}_{i \neq j} \left[ \cos\left(\phi(t_i), \phi(t_j)\right) \right] \tag{1}$$

where $\phi(t_i)$ is one random sample from $\Phi(t_i)$, and the same for $\phi(t_j) \in \Phi(t_j)$. The expectation is taken over all pairs of different types. Similarly, we define the **intra-type** cosine similarity as

$$S_{\text{intra}} \triangleq \mathbb{E}_i \left[ \mathbb{E}_{k \neq l} \left[ \cos\left(\phi_k(t_i), \phi_l(t_i)\right) \right] \right] \tag{2}$$

where the inner expectation is over different embeddings $\phi(t_i)$ for the same type $t_i$, and the outer expectation is over all types. Both $S_{\text{inter}}$ and $S_{\text{intra}}$ take values between $-1$ and $1$. Note that for i.i.d. Gaussian random samples $x, y$, the expected cosine similarity $\mathbb{E}[\cos(x, y)] = 0$. A cosine value closer to 0 often indicates strong isotropy.

Clearly, the inter-type metric describes the similarity between different types, where the intra-type one measures similarity between same type's embedding instances. Our definitions of $S_{\text{inter}}$ and $S_{\text{intra}}$ are similar to the measures used in Ethayarajh (2019), but at the corpus level. Note that some types are more frequent than others, especially under a Zipfian distribution (Piantadosi, 2014), and therefore, the size of $\Phi(t)$ varies dramatically with the frequency of type $t$.

## 2.3 AN INITIAL LOOK AT ANISOTROPY

Inspired by Ethayarajh (2019), we follow their procedure and take a first look at the anisotropy identified by Mimno & Thompson (2017) and Ethayarajh (2019), in the contextual embedding space.

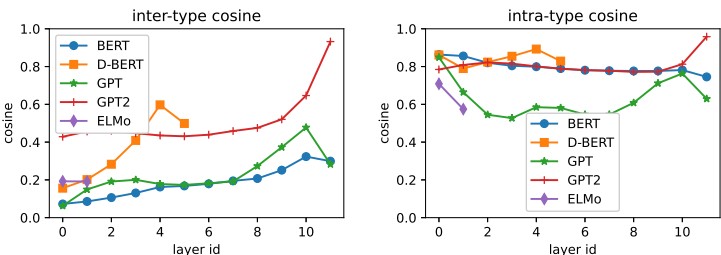

Figure 1: $S_{\text{inter}}$ (left) and $S_{\text{intra}}$ (right). The $S_{\text{inter}}$ increases as layer goes deeper, especially for GPT2's last layer. The $S_{\text{intra}}$ are generally high. This means arbitrary vectors have high cosine similarities.

Figure 1 shows strong anisotropy effects in a number of models. These findings are consistent with Ethayarajh (2019), though we use slightly different metrics. The plots show expected cosine ($S_{\text{inter}}$ and $S_{\text{intra}}$) as a function of layer. For efficiency, we approximate $S_{\text{intra}}$ by imposing a limit of 1,000 samples for frequent types, $t$, if $|\Phi(t)| > 1000$. From the figure we can see the following:

- Both $S_{\text{inter}}$ and $S_{\text{intra}}$ are high ($\gg 0$) across almost all the layers and all the models. In particular, the same as reported in Ethayarajh (2019), GPT2 is relatively more anisotropic.
- $S_{\text{inter}}$ tends to increase with layer, in contrast with $S_{\text{intra}}$ which in general decreases but with fluctuations. This means that embeddings for different types are moving closer to one another at deeper layers, while embeddings for the same type's instances are spreading away.

- The last layer is often special. Note that the last layer has smaller cosines than the second last in most cases, with the notable exception of GPT2.

In summary, we observe large cosines (across layers/models), especially for the GPT2 model. When cosines are close to 1, embeddings lie in a subspace defined by a very narrow cone (Ethayarajh, 2019). One might expect embeddings to be more effective if they took advantage of a larger subspace. Are these models missing an opportunity to have the benefits from isotropy (Mu & Viswanath, 2018)? We answer this question in the following sections.

## 3 CLUSTERS IN THE EMBEDDING SPACE

### 3.1 EFFECTIVE DIMENSIONS

There are $m = 768$ embedding dimensions for BERT, D-BERT, GPT and GPT2, and $m = 1024$ dimensions for ELMo. We perform PCA to reduce the number of dimensions from $m$ down to $k$. For each layer of each model, we start with the data matrix, $M \in \mathcal{R}^{n \times m}$, where $n$ is the number of input tokens ($n = 1.2M$ for PTB dataset), and $m$ is the original number of dimensions. After PCA, we end up with a smaller matrix, $\hat{M} \in \mathcal{R}^{n \times k}$. Let the **explained variance ratio** be: $r_k = \sum_{i=0}^{k-1} \sigma_i / \sum_{i=0}^{m-1} \sigma_i$, where $\sigma_i$ is the $i$-th largest eigen value of $M$'s covariance matrix. In this way, we define the $\epsilon$-**effective-dimension** to be: $d(\epsilon) \triangleq \arg\min_k r_k \geq \epsilon$. For example, $d(0.8) = 2$ means that 2 dimensions capture $80\%$ of the variance. There is a direct connection between $d$ and isotropy: the larger $d$ often implies more isotropy, as data spreads in multiple dimensions.

Table 1: The effective dimension $d(0.8)$

| Layer | 0 | 1 | 2 | 3 | 4 | 5 | 6 | 7 | 8 | 9 | 10 | 11 |
|---|---|---|---|---|---|---|---|---|---|---|---|---|
| BERT | 262 | 273 | 271 | 273 | 276 | 283 | 288 | 282 | 282 | 282 | 283 | 270 |
| D-BERT | 244 | 226 | 232 | 227 | 217 | 175 | | | | | | |
| GPT | 265 | 141 | 65 | 76 | 173 | 210 | 205 | 217 | 221 | 253 | 269 | 307 |
| GPT2 | 114 | 73 | 1 | 1 | 1 | 1 | 1 | 8 | 26 | 66 | 116 | 1 |
| ELMo | 455 | 367 | | | | | | | | | | |

Table 1 reports $d(0.8)$ for different layers and models. It is surprising that GPT2 has so few effective dimensions, especially, $d(0.8) = 1$ for layer 2 to 6. The surprisingly small effective dimensionality is another way of saying that GPT2 vectors fall in a narrow cone, and consequently, their pairwise cosines are large. If all the vectors lie on a 1-D line, all the cosines would be 1, and there would be hardly any model capacity. These observations motivate us to look deeper into the embedding space.

### 3.2 ISOLATED CLUSTERS

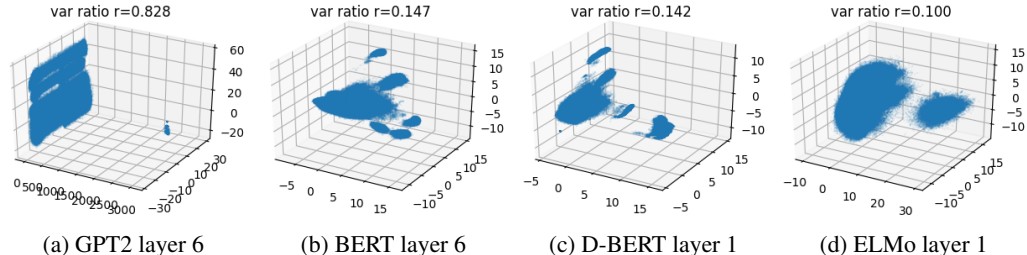

| var ratio r=0.828 | var ratio r=0.147 | var ratio r=0.142 | var ratio r=0.100 |
|---|---|---|---|
| (a) GPT2 layer 6 | (b) BERT layer 6 | (c) D-BERT layer 1 | (d) ELMo layer 1 |

Figure 2: Isolated clusters exist in the embedding spaces for all the models. Here we only show a few representative middle layers for each model. The full visualization can be found in supplementary.

By performing PCA to project the original data into a 3-D view, we can visualize GPT2's layer 6's embedding space in Figure 2a. The three axes refer to the first three principle components, which account for $82.8\%$ of the total variance. All the explained variance ratio will be reported throughout the rest of the paper. The axes values are raw coordinates after PCA. In Figure 2a, there are two disconnected islands that are far away from each other. Note that the first dimension coordinate values

spans from 0 to 3000, significantly wider than the other 2 dimensions. In fact this first principle dimension dominates the total variance. The left island is bigger than the one on the right. The fact that the two islands are so well separated by the first principle component suggests that classifying points by island membership accounts for much of the variance. This two-island property is exhibited in layers 2 through 10 for GPT2. The two islands merge into a single large cluster in the last layer.

We observe similar clustering behavior for all the models across all the layers, though the separations are less distinct, as illustrated in other panels of Figure 2. This is also consistent with Table 1, the less separation, the higher $d(\epsilon)$ values. For GPT2, we had hoped to find that some types are associated with one cluster and other types are associated with the other cluster, but that is not verified in our experiments. Please refer to the supplementary for visualizations of all layers in all the models.

### 3.3 CLUSTERING

Previous literature estimated the space isotropy on pairs of arbitrary tokens, which could reside in two disconnected clusters. But given that the variance is dominated by distances between clusters, such estimation would be biased by the inter-cluster distances. It is more meaningful to consider a per-cluster investigation rather than a global estimate.

We start by performing clustering on the embedding space. There are many methods for clustering. We chose K-Means (https://scikit-learn.org/stable/modules/classes.html#), because it is reasonably fast for large inputs ($n = 1.2$ million vectors) in high ($m \geq 768$) dimensions. DBSCAN algorithm (Ester et al., 1996) could be an alternative as it is density based, but only works on small dataset. We use the Silhouette method (Rousseeuw, 1987) to determine the number of clusters, $|C|$. After running K-means, each point $p$ (one of the $n$ vectors in $M$) is assigned to one of $C$ clusters. For a data point $p$ assigned to the cluster $c \in C$, calculate the following:

$$a_p = \frac{1}{|c|-1} \sum_{q \in c, \; p \neq q} \text{dist}(p,q) \; ; \quad b_p = \min_{\tilde{c} \neq c} \sum_{q \in \tilde{c}} \text{dist}(p,q) \; ; \quad s_p = \begin{cases} \frac{b_p - a_p}{\max(a_p, b_p)}, & \text{if } |c| > 1 \\ 0, & \text{otherwise} \end{cases}$$

where $a_p$ is the mean distance between $p$ and other points in the same cluster; $b_p$ is the minimum (min over $\tilde{c}$) mean distance between $p$ to points of another cluster $\tilde{c}$; and $s_p$ is the Silhouette score for point $p \in c$. The $s_p$ takes value $\in [-1, 1]$. The higher $s_p$, the better assignment of $p$ to its cluster. Better choices of $|C|$ would lead to better values of $s_p$ (and better clustering). We define the **Maximum-Mean-Silhouette** (MMS) score for the embedding space as: $\text{MMS} \triangleq \max_{\text{different } |C|} \mathbb{E}_p[s_p]$, where the maximum is over different $|C|$ values for K-Means. Since it is not feasible to evaluate all choices of $|C| \in [1, n]$, we consider $|C| \in [1, 15]$. The expectation $\mathbb{E}_p[s_p]$ (the mean Silhouette score), is estimated from $20,000$ sample vectors in $M$. We select the best $|C|$ that yields MMS.

The MMS values provide a systematic way to describe how the clusters are distributed in the space. If the clusters are very distinct and splitted, this yields a higher MMS. On the other hand, if clusters are overlapping, blurring together, the MMS score will be low. Note that if MMS $< 0.1$, we set $|C|$ to be 1, as the Silhouette score does not show significant evidence of more clusters.

Table 2: Number of clusters $|C|$

| Layer | BERT | D-BERT | GPT | GPT2 | ELMo |
|-------|------|--------|-----|------|------|
| 0 | 6 | 7 | 1 | 2 | 2 |
| 1 | 6 | 10 | 2 | 2 | 2 |
| 2 | 4 | 15 | 2 | 2 | |
| 3 | 4 | 14 | 2 | 2 | |
| 4 | 3 | 10 | 2 | 2 | |
| 5 | 14 | 2 | 2 | 2 | |
| 6 | 6 | | 2 | 2 | |
| 7 | 2 | | 2 | 2 | |
| 8 | 2 | | 2 | 2 | |
| 9 | 11 | | 1 | 2 | |
| 10 | 2 | | 1 | 2 | |
| 11 | 9 | | 1 | 2 | |

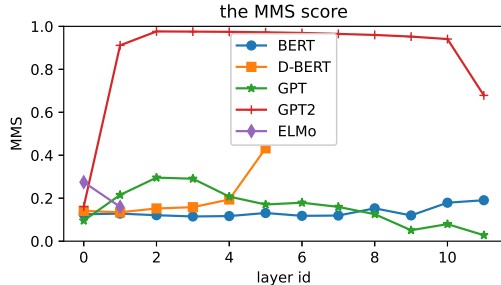

Figure 3: The MMS for all the models. GPT2 has significantly higher MMS scores than other models from layer 1 to layer 11. This means the cluster effects are more severe in GPT2.

Table 2 makes it clear that clustering plays an important role in most layers of most models. Some models (BERT and D-BERT) have more clusters, and some have fewer (GPT, GPT2, ELMo). This dichotomy of models is also reflected in Figure 2.

Maximum-Mean-Silhouette scores are shown in Figure 3. There are significantly higher MMS values for GPT2, starting from the 2nd layer. Recall in Figure 2a, we showed that two far-away islands exist in the space and their distance dominates the variance. In Figure 3, the high MMS scores also verifies that. Another interesting observation is, for causal models GPT, GPT2 and ELMo, they all have higher MMS in their middle layers but lower MMS in the end. This means their initial layer and final layers' embeddings tend to merge. On the contrary, the BERT and DistilBERT have increasing MMS in deeper layers, meaning that the clusters in their embeddings are becoming clearer in deeper layers.

### 3.4 Isotropy in centered space within clusters

As suggest by Mu & Viswanath (2018), the embedding space should be measured after shifting the mean to the origin. We subtract the mean for each cluster, and calculate the adjusted $S_{\text{inter}}$. Assuming we have a total of $|C|$ clusters, let $\Phi^c(t) = \{\phi_1^c(t), \phi_2^c(t), \ldots\}$ be the set of type $t$'s embeddings in cluster $c \in C$, and $\phi^c(t)$ be one random sample in $\Phi^c(t)$. Define the adjusted similarity:

$$S'_{\text{inter}} \triangleq \mathbb{E}_c \left[ \mathbb{E}_{i \neq j} \left[ \cos \left( \bar{\phi}^c(t_i), \bar{\phi}^c(t_j) \right) \right] \right] \quad , \quad \text{where} \quad \bar{\phi}^c(t) = \phi^c(t) - \mathbb{E}_{\phi^c} \left[ \phi^c(t) \right] \quad (3)$$

Here $\mathbb{E}_c$ is the average over different clusters, and $\bar{\phi}^c(t)$ is the original embedding shifted by mean (subtract the mean), where the mean is taken over the samples in cluster $c$. Similarly we define

$$S'_{\text{intra}} \triangleq \mathbb{E}_c \left[ \mathbb{E}_i \left[ \mathbb{E}_{k \neq l} \left[ \cos \left( \bar{\phi}_k^c(t_i), \bar{\phi}_l^c(t_i) \right) \right] \right] \right] \quad (4)$$

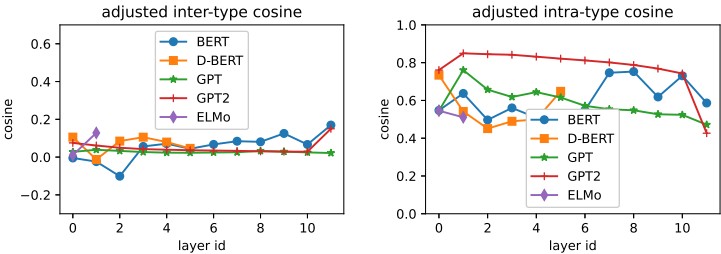

Figure 4: $S'_{\text{inter}}$ (left) and $S'_{\text{intra}}$ (right). The adjusted $S'_{\text{inter}}$ are close to zero, meaning that the space is isotropic under the adjusted measure.

The Figure 4 illustrates the adjusted cosine similarities $S'_{\text{inter}}$ and $S'_{\text{intra}}$. It reveals that:

- For the adjusted inter-type cosine (the left plot), all models are having consistent near-zero $S'_{\text{inter}}$. This means nearly perfect isotropy exists within each cluster, in each layer of all the models. The last layer of GPT2 and BERT has slightly worse isotropic behavior, nevertheless, general inter-type isotropy stays across all layers. This reveals the distinguishable embedding vectors.
- The general decreasing trend of intra-type cosine (the right plot) shows that the multiple instances for the same type/word, is slowly spreading over the layers. This is consistent with the un-centered intra-type cosine shown in Figure 1.

## 4 Low-dimensional Manifolds

### 4.1 Swiss Roll manifold of GPT/GPT2

While BERT and D-BERT tend to distribute embeddings along more dimensions, GPT and GPT2 embed tokens in low-dimensional manifolds in their contextual embedding spaces. More specifically, we discover that most of the tokens are embedded on a spiral band, and that band gets thicker in the later layers thereafter form a Swiss Roll shaped surface.

Figure 5a and 5b show the 2-D front view of the manifold in GPT and GPT2. Figure 5a zooms into the large cluster illustrated in Figure 2a (the left one), and discards the smaller one (the right one).

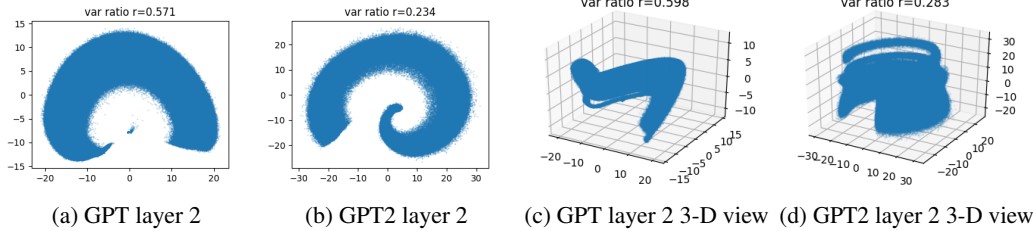

Figure 5: The 2-D and 3-D view of low-dimensional manifold in GPT/GPT2's embedding spaces

3-D plots are shown in Figure 5c and 5d to demonstrate two manifolds, a band shaped manifold and a Swiss Roll shaped manifold. These plots were computed over PTB dataset. Similar results have been obtained from WikiText-2 in supplementary. Figure 6 tracks the progression of a narrow band into a Swiss Roll. The Swiss Roll becomes taller and taller with deeper and deeper layers.

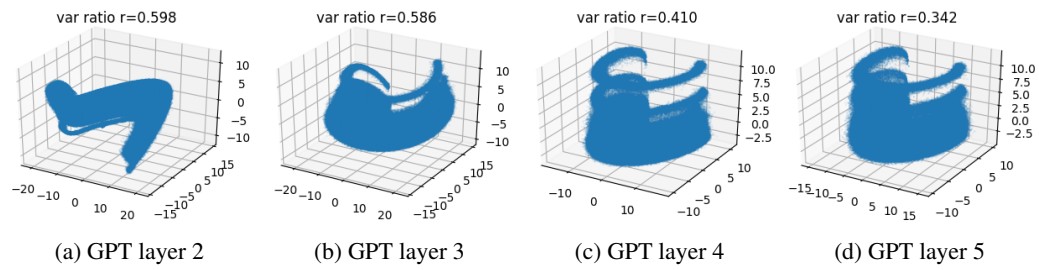

Figure 6: The evolution from a narrow band into a taller and taller Swiss Roll with deeper layers.

## 4.2 TOKENS IN THE SPACE

To verify the manifold structure in GPT family, we study the token embeddings in the space. It is believed that similar embeddings (e.g. the embeddings for two instances of the same word) tend to stay close together in a Euclidean space, as they should have high cosine similarities. Figure 7 drills down into the embeddings for six frequent words: three punctuation symbols ("\", "&", ".") and three common words ("the", "first", "man"). Each panel uses four colors: three colors (black, red, green) for three words of interest, plus gold color for all the other tokens.

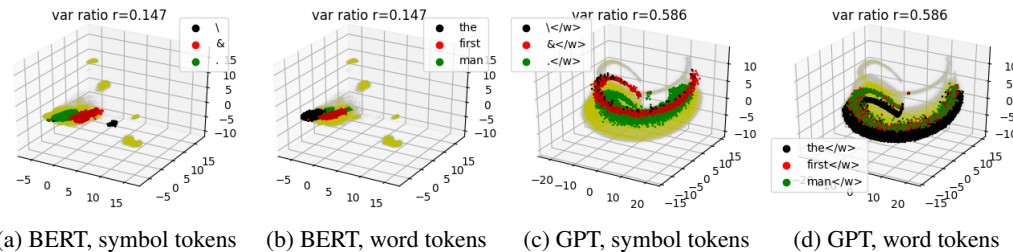

Figure 7: Embeddings for symbol tokens and word tokens, in layer 3 of BERT and GPT. This shows that GPT has manifold structure, such that vectors are along the spiral band. BERT's space is closer to a Euclidean space as similar vectors are in concentrated clusters.

As shown in Figure 7a 7b, the BERT model indeed group similar embeddings into small regions in the space (the red, black and green clusters). However, the GPT models are assigning similar embeddings along the manifold we observed before. In Figure 7c 7d, the embeddings for the tokens occupy a spiral band that almost cross the entire space. It does not comply with the Euclidean space geometry as points in such a spiral band would not have high cosine similarity. A Riemannian metric must exist, such that the manifold has larger distance between two spiral bands, but smaller distance on the

band. Note that the 3-D plots are obtained using PCA, so there is no density-based nor non-linear reduction involved. Therefore, the manifold structures in GPT embedding spaces are verified.

## 4.3 WORD FREQUENCY

Another key finding is that all the models are trying to map the high frequent words/types to some specific region in the embedding spaces, rather than spreading them out to the entire space. In Figure 8, embeddings ( 8a 8c ) and corresponding word frequencies ( 8b 8d ) of GPT's layer 8 and 9 are shown. The darker red denoted higher frequency and blue is lower frequency. The numbers at the colorbar show the number of occurrence (of a particular word / type).

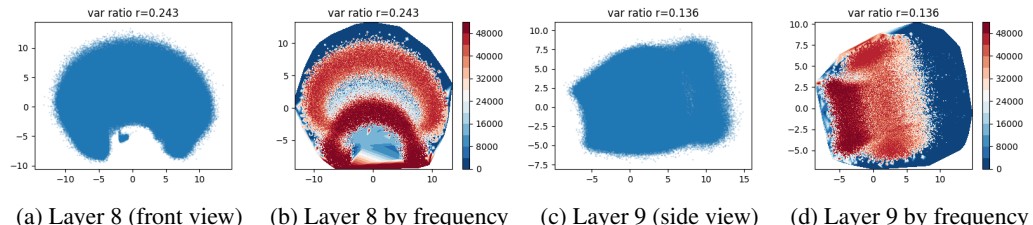

(a) Layer 8 (front view)  (b) Layer 8 by frequency  (c) Layer 9 (side view)  (d) Layer 9 by frequency

Figure 8: Word frequency heatmap in GPT layer 8 and 9. Red is high frequency, blue is low. High frequency words are at the front end of the Swiss Roll, while low frequency words at the other end. ( 8b 8d are drawn using matplotlib.tricontourf, so the ring at 8b's bottom should not be closed.)

The Figure 8a 8c are after PCA, and selecting the two most significant dimensions. From GPT layer 8 to layer 9, as the Swiss Roll becomes taller, more variance is accounted for along the height of the Swiss Roll. Thus, the perspective switches from a front view to a side view when moving to layer 9.

Figures 8b and 8d show that the most frequent words appear at the head of the Swiss Roll, followed by bands of less and less frequent words. The least frequent words appear at the far end of the Swiss Roll. This pattern suggests the model distinguishes more frequent from less frequent words. As the model finds more and more rare words, it appends them at the end of the Swiss Roll.

## 4.4 MANIFOLD LOCAL INTRINSIC DIMENSION

Although the original space dimension is 768 (1024 for ELMo), the manifold we observed has a lower **intrinsic dimension**. It means the data point on the manifold has fewer degrees of freedom to move around. For example, on a Swiss Roll in a 3-D space, any point can only have 2-D freedom thus the intrinsic dimension is only 2. A recent research on the intrinsic dimension for deep networks could be found at (Ansuini et al., 2019). In this section, we adopt the **Local Intrinsic Dimension (LID)** that estimates dimension locally with respect to a reference point. LID is introduced by Houle (2013), and being used in deep learning model characterization recently, e.g. (Ma et al., 2018). The LID is often derived using expansion models (Houle et al., 2012), which tries to obtain the local dimension in the vicinity of a reference point from the growth (expansion) characteristics. To illustrate this, we borrow an example from Ma et al. (2018). Let $\gamma$ be the radius of an $m$-D ball in the Euclidean space, denote its volume as $\nu$, then the volume's growth rate is proportional to $\gamma^m$, i.e. $\nu_2/\nu_1 = (\gamma_2/\gamma_1)^m$, from which we can infer the local dimension $\tilde{m}$ by $\tilde{m} = \log(\nu_2/\nu_1) \,/\, \log(\gamma_2/\gamma_1)$.

Accurately computing LID is a hard problem which requires a tremendous amount of data samples and enough density around the reference point. So fewer-sample estimate of LID is being studied in the past decade. One of the efficient estimation is proposed by Amsaleg et al. (2015). This technique relies on $K$ nearest neighbor search ($K$-NN). For a reference point $p$, denote the set of its $K$ nearest neighbor points as $\Psi_p = \{q_1, \ldots, q_K\}$. Then the estimate of LID is computed as: $\tilde{\text{LID}}(p) = -\left(\frac{1}{K}\sum_{i=1}^{K}\log\frac{\text{dist}(p,q_i)}{\max_i(\text{dist}(p,q_i))}\right)^{-1}$, where the term inside $\log$ is the ratio of distance between $p$ to its neighbor, over the maximum distance among them. In our analysis, we use an efficient nearest neighbor computation package FAISS (Johnson et al., 2017) (https://github.com/facebookresearch/faiss) to perform the $K$-NN. We set $K = 100$, the same as in (Aumüller & Ceccarello, 2019). $\ell_2$ distance is used, i.e. $\text{dist}(p,q) = \|p - q\|_2$. We report the mean LID over all the samples $p$, as $\mathbb{E}_p[\tilde{\text{LID}}(p)]$, in Figure 9.

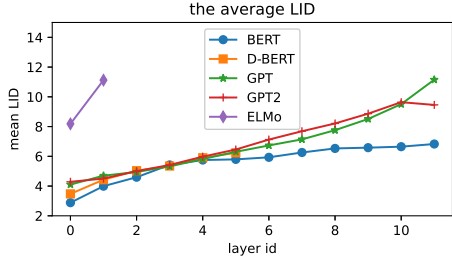

Figure 9: The average LID using Euclidean distance. ELMo's original embdding dimension is 1024, larger than other models' 768.

Table 3: A comparison of LIDs (using cosine similarity) among contextual and static embedding spaces.

| | Model | $n$ | $m$ | avg LID |
|---|---|---|---|---|
| Contxt Embeds | BERT | 1.19 M | 768 | 5.6 |
| | D-BERT | 1.19 M | 768 | 7.3 |
| | GPT | 0.96 M | 768 | 6.8 |
| | GPT2 | 1.09 M | 768 | 7.0 |
| | ELMo | 0.88 M | 1024 | 9.1 |
| Static Embeds | GloVe | 1.18 M | 100 | 18.0 |
| | GloVe-2M | 2.20 M | 300 | 26.1 |
| | GNEWS | 3.00 M | 300 | 21.1 |

As shown in Figure 9, the mean LIDs for all the models in all the layers are below 12. The small mean LID values reveals that the manifold's intrinsic dimension is relatively low, especially considering that this is a 768-D (1024 for ELMo) embedding space. Since ELMo's 1024-D is larger than other models 768-D dimension, its LID is also slightly higher than other models as shown in the figure. The existence of a low-dimensional embedding is also suggested in (Reif et al., 2019) when they study the BERT embedding geometry.

In all the contextual embedding layers, there is a clear trend of increasing LID values. In Figure 9, we can also see a nearly-linear relationship between layer id and LID. With deeper and deeper layers in the net, the manifold is diffusing and slowly loses concentration. This would lead to data samples spreading, consistent with Figure 4 (recall that intra-type cosines decrease with depth). Note that as layer goes deeper, each token embedding is collecting information from context by adding their embeddings (and non-linear transforms concatenated). This could explain the spreading / expanding of the local subspace, and therefore the LID increases in deeper layers.

Table 3 compares LIDs for static and contextual embeddings. The table reports results for three static embeddings: GloVe / GloVe-2M (Pennington et al., 2014), and GNEWS (Mikolov et al., 2013a). Results for static embedding LIDs are based on Aumüller & Ceccarello (2019). Following Aumüller & Ceccarello (2019), we use cosine distance here: $\text{dist}'(p,q) = 1 - \cos(p,q) = 1 - \frac{\langle p,q \rangle}{\|p\|_2 \|q\|_2}$. Note that estimates for LID using cosines are very close to the estimates using $\ell_2$ distances. Table 3 reports averages of LIDs over each model's layers. Even though GloVe (Pennington et al., 2014) in Table 3 has much fewer embedding dimensions (100-D compared with BERT's 768-D), the LID is still higher than all of the contextual embedding models. From the table we can find that static embedding spaces generally have higher LID than the contextual ones. This means that the data points are more isotropic in the static embeddings, possibly due to their large vocabularies.

## 5 CONCLUSIONS AND FUTURE WORK

Previous works have reported the strong anisotropy in deep LMs, which is hard to explain the superior performance achieved by these models. We suggest that the anisotropy is a global view, being largely misled by distinct clusters resided in the space. Our analysis show that it is more constructive to isolate and transform the space to measure the isotropy. From this view, within the clusters, the spaces of different models all have nearly perfect isotropy that could explain the large model capacity. In addition, we investigate the space geometry for different models. Our visualization demonstrates a low-dimensional Swiss Roll manifold for GPT and GPT2 embeddings, that has not been reported before. The tokens and word frequencies are presented to qualitatively show the manifold structure. We propose to use the approximate LID to quantitatively measure the local subspace, and compared with static embedding spaces. The results show smaller LID values for the contextual embedding models, which can be seen as a local anisotropy in the space. We hope this line of research could bring a comprehensive geometric view of contextual embedding space, and gain insights on how the embeddings are affected by attention, compression, multilingualism, etc. Therefore the model performance could be further improved based on the findings.

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

SUPPLEMENTARY: FULL RESULTS ON PTB AND WIKITEXT-2 DATASETS

## A    RESULTS ON WIKITEXT-2 DATASET

### A.1    THE UNADJUSTED INTER AND INTRA COSINE SIMILARITY

Note that "dist" in the following legends represents DistilBERT model.

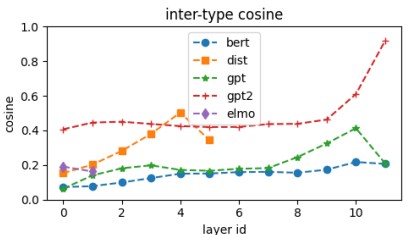 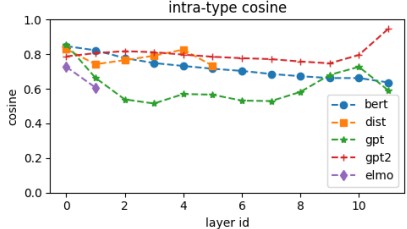

(a) Inter-type cosine similarity. As layers goes deeper, inter-type cosine goes higher. All models' last layer behaves slightly differently.

(b) Intra-type cosine similarity. The intra-type cosine decreases showing the same type's embedding instances are spreading in deeper layers.

### A.2    THE CENTER-SHIFTED AND CLUSTERED COSINE SIMILARITY

The inter-type and intra-type cosines are adjusted using the proposed center-shifting and clustering methods. Now it reflects the isotropy in almost all layers in all models.

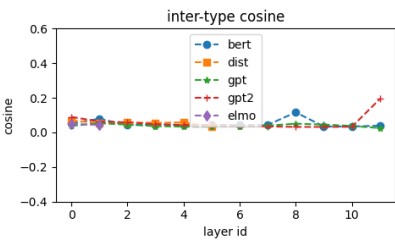 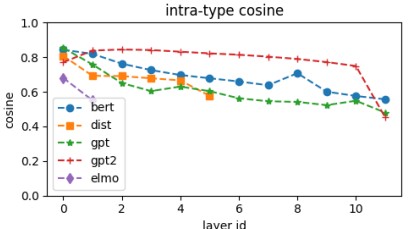

(a) The clustered inter-type cosine (center shifted). It shows strong isotropy as the average cosine between different types is close to 0 across all layers in all models. The GPT2's last layer still has slighly higher cosine compared with others.

(b) The clustered intra-type cosine (center shifted). The intra-type cosines are much more consistent than the unadjusted counterpart (the cosine decreases nearly monotonically as layers goes deeper).

### A.3    THE APPROXIMATE LOCAL INTRINSIC DIMENSIONS

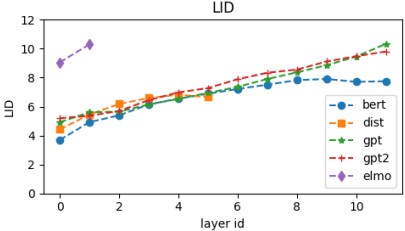

Figure 12: Local Intrinsic Dimensions. The LID increases as layer goes deeper, reflecting embeddings spreading out in all models' deeper layers (becoming more locally isotropic).

# B FULL VISUALIZATION - PTB DATASET

## B.1 BERT

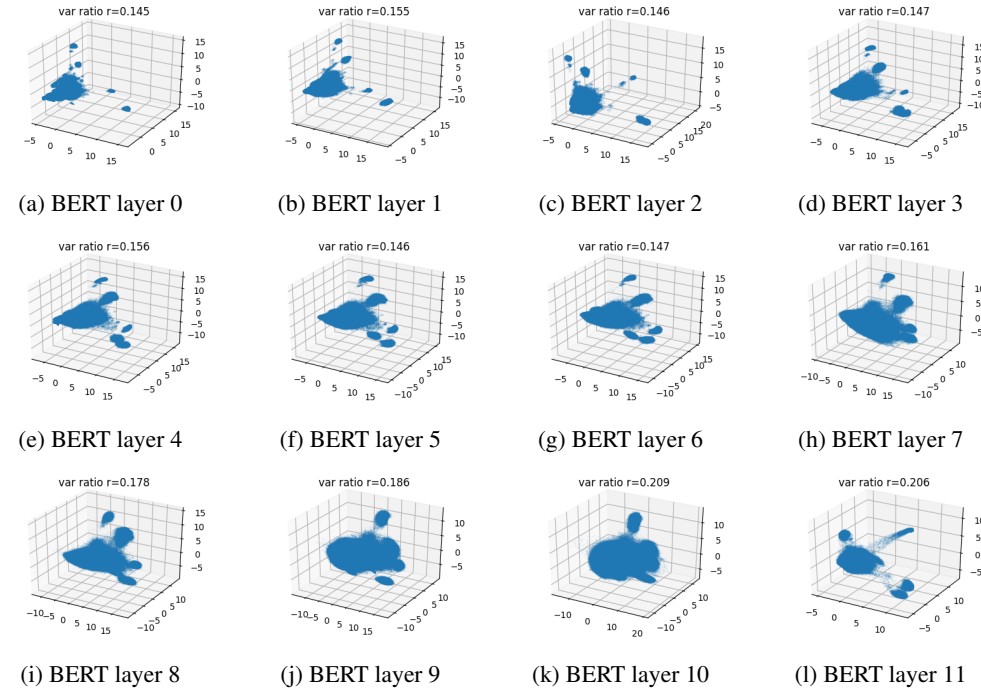

(a) BERT layer 0    (b) BERT layer 1    (c) BERT layer 2    (d) BERT layer 3

(e) BERT layer 4    (f) BERT layer 5    (g) BERT layer 6    (h) BERT layer 7

(i) BERT layer 8    (j) BERT layer 9    (k) BERT layer 10    (l) BERT layer 11

## B.2 DISTILBERT AND ELMO

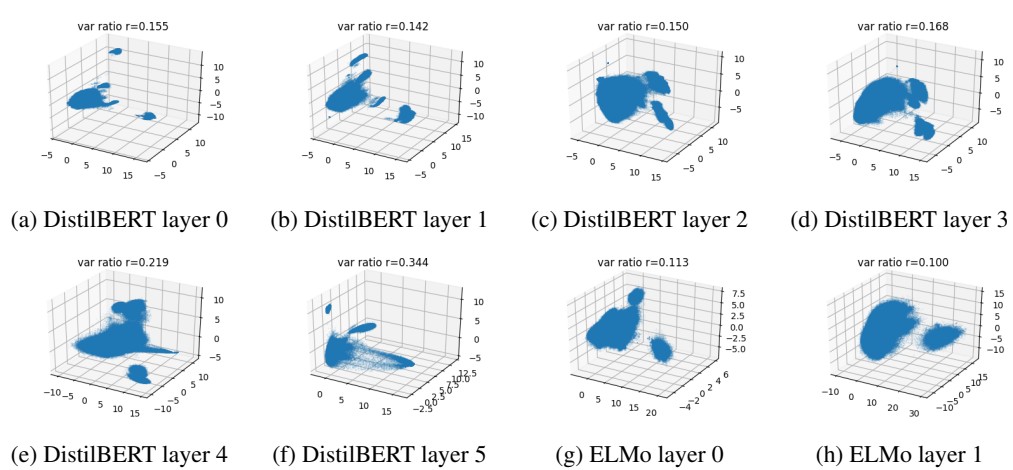

(a) DistilBERT layer 0    (b) DistilBERT layer 1    (c) DistilBERT layer 2    (d) DistilBERT layer 3

(e) DistilBERT layer 4    (f) DistilBERT layer 5    (g) ELMo layer 0    (h) ELMo layer 1

### B.3 GPT

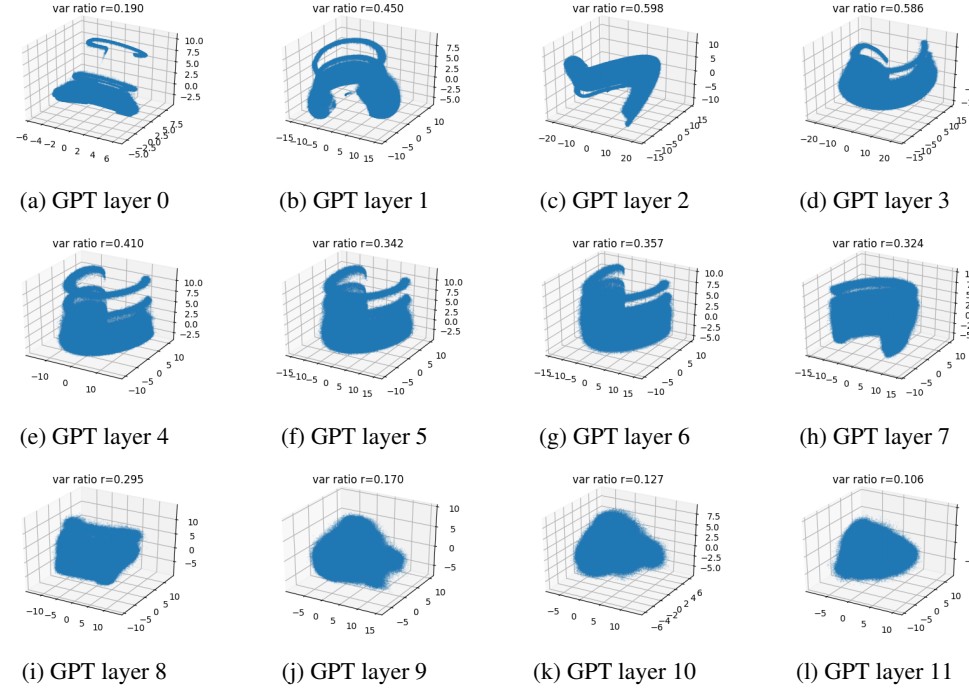

(a) GPT layer 0     (b) GPT layer 1     (c) GPT layer 2     (d) GPT layer 3

(e) GPT layer 4     (f) GPT layer 5     (g) GPT layer 6     (h) GPT layer 7

(i) GPT layer 8     (j) GPT layer 9     (k) GPT layer 10     (l) GPT layer 11

### B.4 GPT2

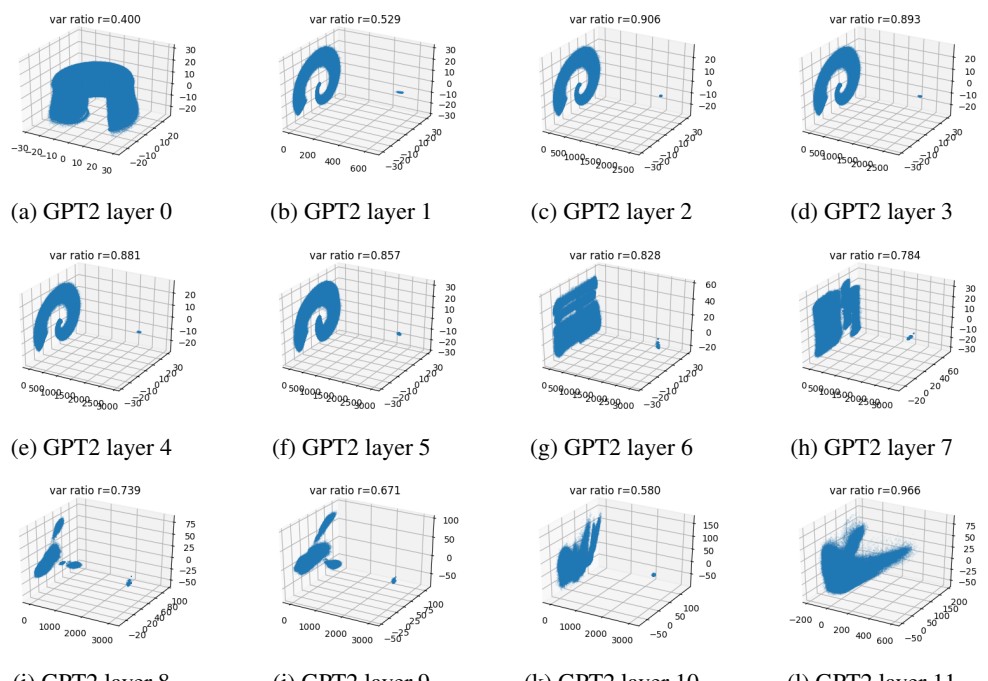

(a) GPT2 layer 0     (b) GPT2 layer 1     (c) GPT2 layer 2     (d) GPT2 layer 3

(e) GPT2 layer 4     (f) GPT2 layer 5     (g) GPT2 layer 6     (h) GPT2 layer 7

(i) GPT2 layer 8     (j) GPT2 layer 9     (k) GPT2 layer 10     (l) GPT2 layer 11

# C  Full Visualization - WikiText-2 Dataset

## C.1  BERT

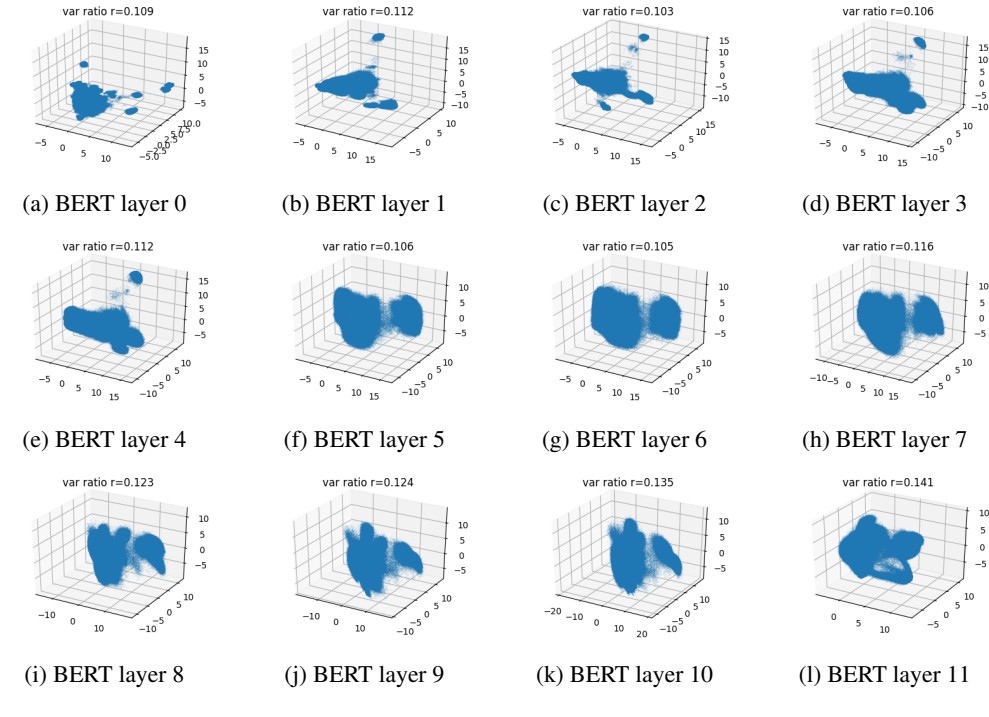

(a) BERT layer 0    (b) BERT layer 1    (c) BERT layer 2    (d) BERT layer 3

(e) BERT layer 4    (f) BERT layer 5    (g) BERT layer 6    (h) BERT layer 7

(i) BERT layer 8    (j) BERT layer 9    (k) BERT layer 10    (l) BERT layer 11

## C.2  DistilBERT

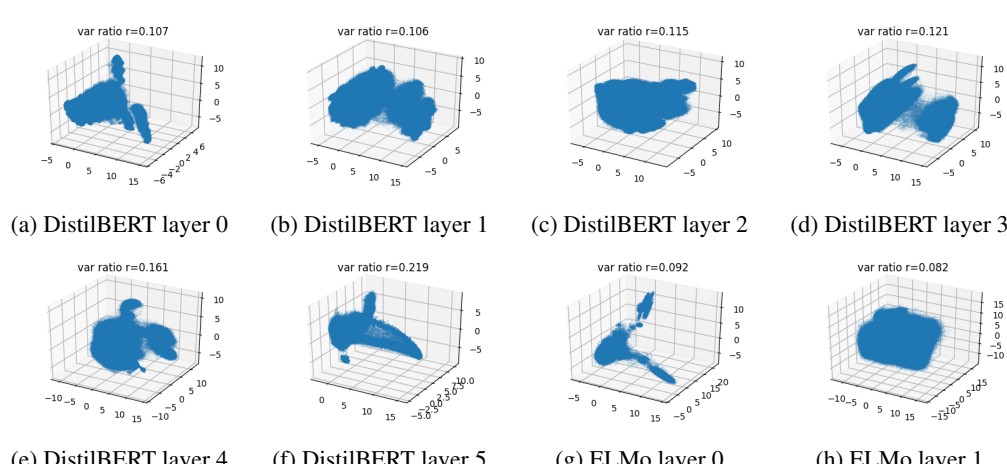

(a) DistilBERT layer 0    (b) DistilBERT layer 1    (c) DistilBERT layer 2    (d) DistilBERT layer 3

(e) DistilBERT layer 4    (f) DistilBERT layer 5    (g) ELMo layer 0    (h) ELMo layer 1

## C.3 GPT

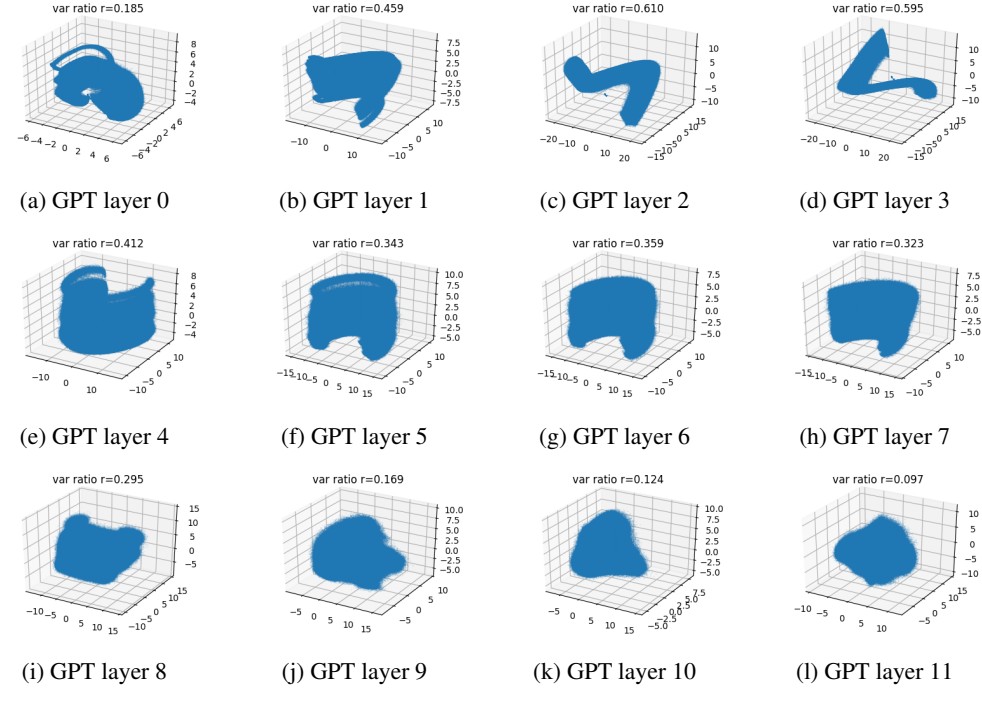

(a) GPT layer 0      (b) GPT layer 1      (c) GPT layer 2      (d) GPT layer 3

(e) GPT layer 4      (f) GPT layer 5      (g) GPT layer 6      (h) GPT layer 7

(i) GPT layer 8      (j) GPT layer 9      (k) GPT layer 10      (l) GPT layer 11

## C.4 GPT2

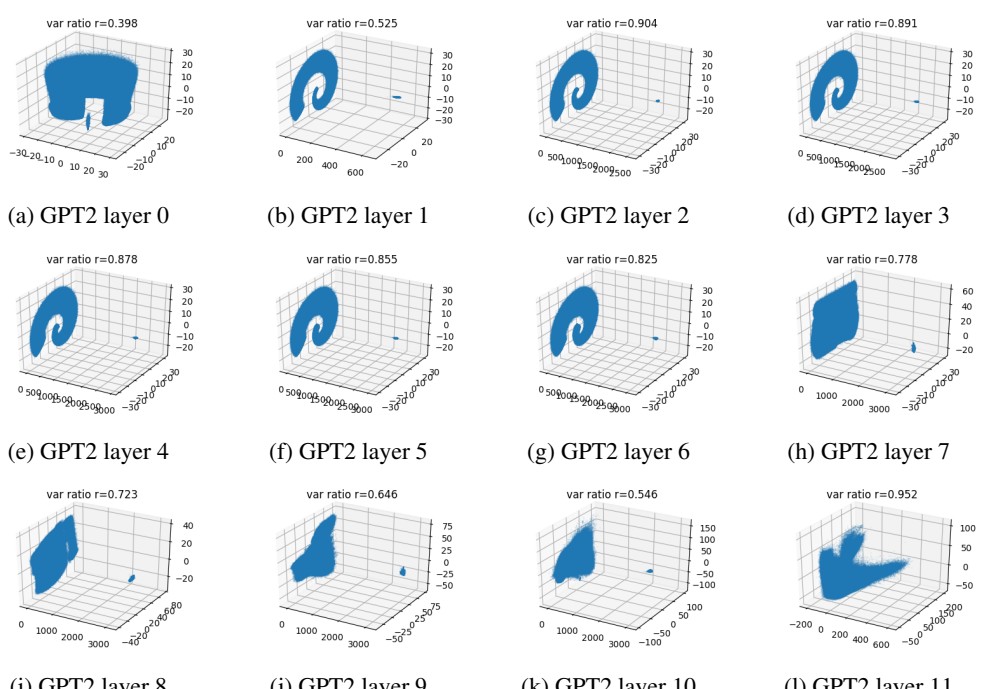

(a) GPT2 layer 0      (b) GPT2 layer 1      (c) GPT2 layer 2      (d) GPT2 layer 3

(e) GPT2 layer 4      (f) GPT2 layer 5      (g) GPT2 layer 6      (h) GPT2 layer 7

(i) GPT2 layer 8      (j) GPT2 layer 9      (k) GPT2 layer 10      (l) GPT2 layer 11

# D  ADDITIONAL STUDIES

## D.1  K-MEANS CLUSTERING ACCURACY

We use K-Means to perform clustering, which raises two issues here. First, K-Means is very sensitive to initialization, different initialization could leads to different clustering results. However, note that in our task, we are not seeking for optimal clustering. Sub-optimal, e.g. treating two overlapping clusters as a big one, is totally fine.

To illustrate this, we add another metric, Davies-Boulding (DB) index (Davies & Bouldin, 1979), to show that slightly different $K$ is fine. This DB index is the average similarity between each cluster and its closest cluster. The value closer to 0, the better clustering is done. We still search in $[2, 15]$, and choose $K$ with the minimum DB index (MDB). MDB sometimes gives different $K$ than that by MMS metric. If MDB is $> 4$, we discard MDB and treat all data as one single cluster. We provide the comparison of selecting $K$ using MMS (left) and MDB (right) here in Table 4. We can see that for less-distinct clusters, e.g. in BERT, two metric could yield different $K$ values, due to merging or splitting. For very separated clusters, e.g. in GPT2, the two metric agrees. We plot the cosines using MDB's $K$ values, in Figure 21. It is similar to Figure 4, which uses slightly different $K$ from MMS. The values are close to 0 indicating isotropy in the center-shifted clusters. This means that the procedure to reveal isotropy, is not sensitive to $K$ in K-Means.

Table 4: $K$ by MMS(left) vs MDB(right)

| Layer | BERT | | D-BERT | | GPT | | GPT2 | | ELMo | |
|---|---|---|---|---|---|---|---|---|---|---|
| 0 | 6 | 14 | 7 | 9 | 1 | 8 | 2 | 5 | 2 | 1 |
| 1 | 6 | 6 | 10 | 15 | 2 | 2 | 2 | 2 | 2 | 1 |
| 2 | 4 | 12 | 15 | 5 | 2 | 2 | 2 | 2 | | |
| 3 | 4 | 15 | 14 | 11 | 2 | 2 | 2 | 2 | | |
| 4 | 3 | 14 | 10 | 2 | 2 | 2 | 2 | 2 | | |
| 5 | 14 | 13 | 2 | 4 | 2 | 2 | 2 | 2 | | |
| 6 | 6 | 14 | | | 2 | 2 | 2 | 2 | | |
| 7 | 2 | 15 | | | 2 | 2 | 2 | 2 | | |
| 8 | 2 | 7 | | | 2 | 2 | 2 | 2 | | |
| 9 | 11 | 6 | | | 1 | 2 | 2 | 2 | | |
| 10 | 2 | 4 | | | 1 | 10 | 2 | 2 | | |
| 11 | 9 | 3 | | | 1 | 15 | 2 | 2 | | |

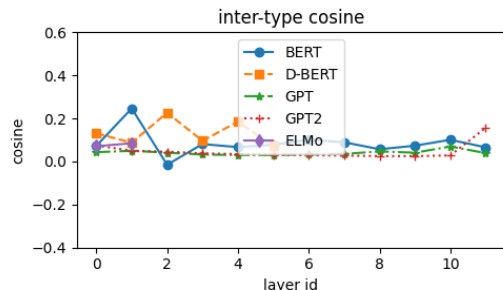

Figure 21: The adjusted inter-type cosines, computed using $K$ from the criteria of minimizing DB index. The values are still close to 0.

Another issue is that K-Means implicitly assumes convex clusters, which often does not hold. In fact, it assumes isotropic convex clusters because we simply use $\ell_2$ distance. However, density-based clustering such as DBSCAN, is too slow thus cannot handle these datasets (million level). This is a trade-off to use K-Means, and empirical results above show that it is efficient and very useful to distinguish separated clusters.

## D.2  CLUSTERS AND WORDS

We study the tokens and their relationship to the clusters existed in the contextual embedding spaces. We picked some representative tokens to see how they are distributed. We also study the very unique small cluster in GPT2, and how it connects to the main cluster that is far away. We obtain the following observations:

- For BERT, high frequent words (e.g. 'the') stays at one side of the main big cluster, while low-frequent words are at the other side.
- For BERT, punctuation are random but occupy distinct islands: '!' is a small cluster close to the main island; '‘' and 'áre distinct islands far away; '?' and some others are on the main cluster.
- For GPT2, almost all single letters (a to z) and mid-to-high frequent would occupy both the left (big) and right (small) islands.
- For GPT2, we didn't find any token that only appears in the right small island. It seems the token in the small island always has mirrors in the left big cluster.

- For word types, e.g. noun, verb, etc, we didn't find a clear pattern. We suspect word frequency affects more than categories.

We provide a few examples. Figure 22, 23 show BERT layer 3, Figure 24, 25 show GPT2 layer 3.

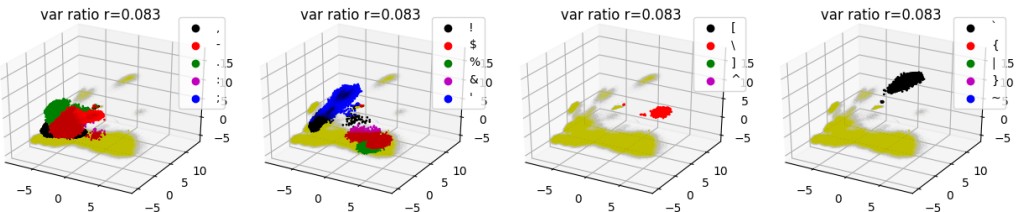

(a) Punctuation are random but less concentrated. They also occupy distinct islands.

Figure 22: BERT Layer 3 Punctuation

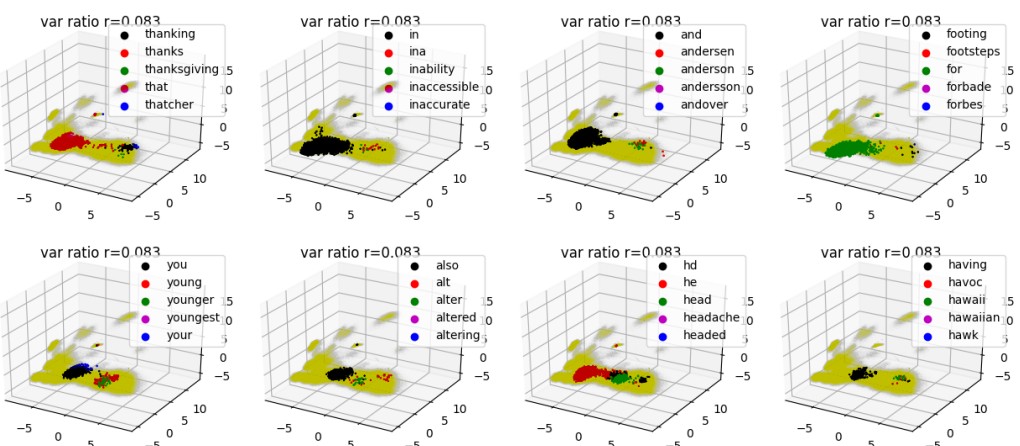

(a) Frequent words and infrequent words are on the main cluster, but at two sides. An evidence that words are distributed based on the frequency.

Figure 23: BERT Layer 3 Words

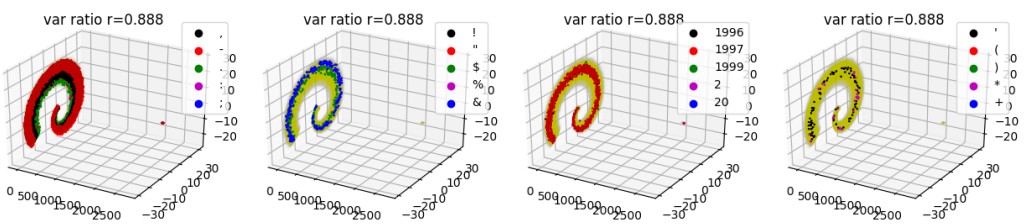

(a) Punctuation are random. Some occupy both islands, some do not.

Figure 24: GPT2 Layer 3 Punctuation

Based on these observations, we have concluded that frequency plays an important role in the token distributions. High frequent words and low frequent words are often taking opposite sides of the space. This is also revealed in Section 4.3. We are yet not clear what causes this, but we suspect it is related to the training process. During training, high frequent words are updated more times. Also, since they are used in many many different context, they play a role as some shared embedding across context. Similar to the XLM model, the shared embedding are often more isotropic and more concentrated. However, this is early hypothesis and due to future research.

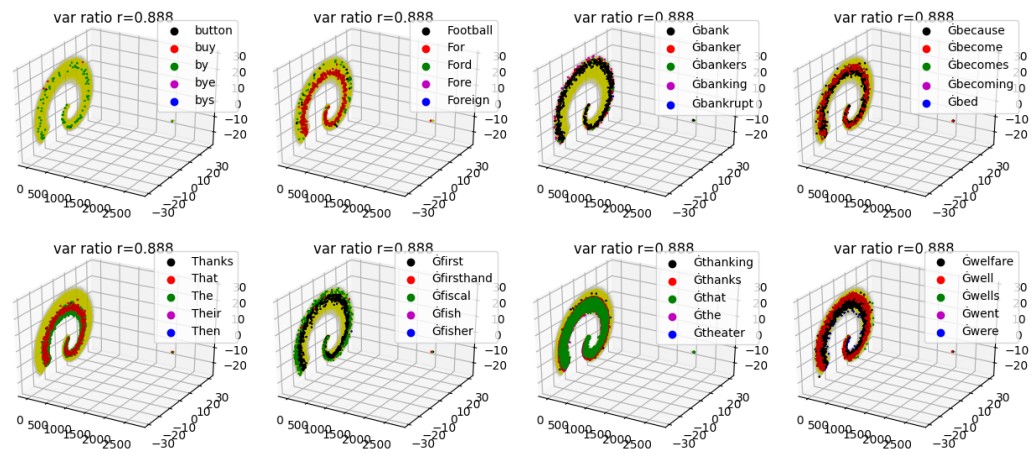

(a) Mid-to-high frequent words often occupy both distinct islands (notice that the right small cluster is also colored), where a roll-shaped alignment can be observed on the larger island.

Figure 25: GPT2 Layer 3 Words

### D.3 EMBEDDING OF TRANSLATION LANGUAGE MODEL XLM

We also perform analysis and visualization on the XLM model (Conneau & Lample, 2019). BERT is mask language model (MLM), GPT is causal language model (CLM), and XLM is translation language model (TLM). We provide visualization of XLM's 6 layers embeddings here. This is on WikiText-2 dataset.

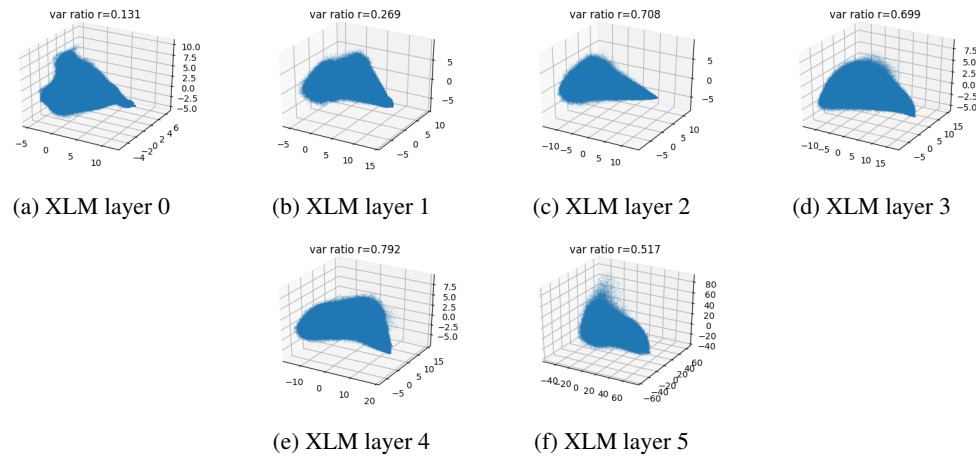

We try to establish a systematic view of embedding geometry for different types of deep LMs. We have hypothesis and very preliminary results here. BERT (an MLM) show spreading clusters, but not very distinct. GPT (an CLM) shows highly separated clusters. XLM (an TLM) does not demonstrate clustering effect, and the embedding are centered.

One possible explanation for XLM's behavior, is that this is a multi-lingual model, and the embedding space have to be shared between languages. This is forced during the training process of this translation language models. In that case, a single cluster residing in the center, would be a good shared embedding across languages. However, this is just hypothesis and requires further study on more models.

## D.4 LID ESTIMATION ROBUSTNESS

We follow (Aumüller & Ceccarello, 2019) to choose $K = 100$ for K-Nearest-Neighbor (K-NN) search for LID approximation, and make a direct comparison with them. It raises the concern that 100 samples might not be enough to effectively estimate the local dimension. We conduct additional experiments here to select $K = 200, 500, 1000$, and demonstrate that the LID estimation is robust. They provide similar LID estimates across all layers, in all the models. Though using more samples indeed obtain very slightly higher values of LID (in Figure 27, we can see a little bit up-shifting from left-most plot to the right-most plot). This is expected, as less number of samples often tends to under-estimate, and over-smoothing of LID. Nevertheless, the LID is still much smaller than the original dimension 768, so using 100 samples is a good trade-off to efficiently approximate LID.

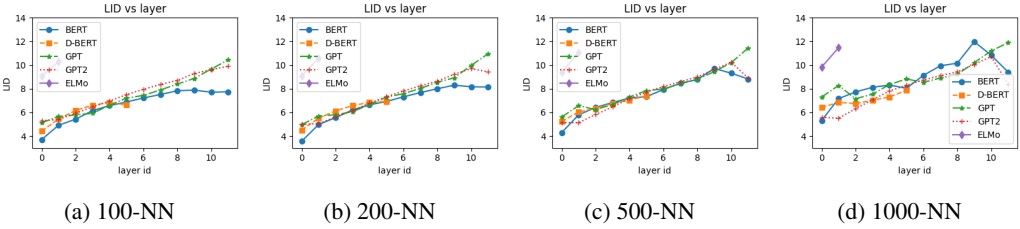

| (a) 100-NN | (b) 200-NN | (c) 500-NN | (d) 1000-NN |

Figure 27: LID estimate using different number of samples for nearest neighbor search.

As layer goes deeper, the LID increases. In other words, the local space dimension expands, at a cost of losing density. For example, the spiral band (1-D) in GPT's front layer, becomes a Swiss Roll (2-D), and the roll surface get thickness (3-D), as layer increases. But we are not clear about the reasons yet, only suspect that data is spreading as more context info is added in later layers (the embedding for a token in deeper layer is based on summation of all embeddings in the context, due to attention). This is due to future study.

## D.5 ABLATION ANALYSIS ON CLUSTERING

To better study the clustering effect, we conduct experiment that computes the inter-type cosines, on clusterd-only embeddings and clustered plus center-shifted embeddings. The following figure shows GPT2's cosine on original embeddings without adjustment (blue), the clustering-only embeddings (orange), and full (clustering + centering) adjustment (green).

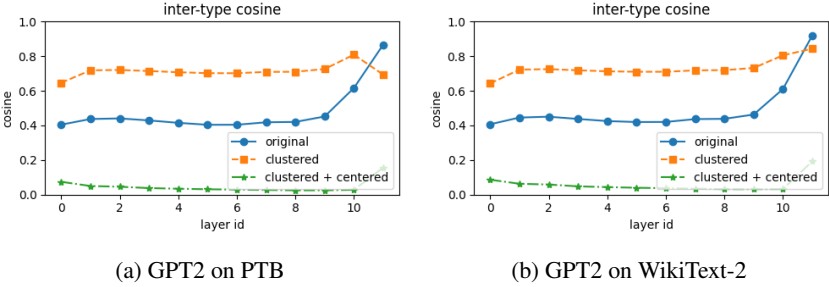

| (a) GPT2 on PTB | (b) GPT2 on WikiText-2 |

In the original embedding without adjustment, we see inconsistent behavior in the last layers. However, if we perform clustering and measure the $S_{\text{inter}}$ within the clusters (orange), we can see much more consistent behavior across layers (more flat curve). This indicates that the clustering effect exists in all the layers, which is also verified by layer-wise visualization in Appendix B.4 and C.4.

Meanwhile, the large values of cosines in the orange curve are expected. Now cosines are only computed within clusters, where those clusters are not at the origin. The higher values here, the more concentrated clusters are. These indicate that after clustering, the subspace within each cluster are now consistent, across all the layers. Finally, we shift those clusters to the origin, and get the green curve (values near 0), indicating isotropic cluster shapes.

### D.6 Positional Encoding in the Geometry

It is very interesting to investigate whether the positional encoding affects the geometry in the contextual embedding spaces. In particular, since GPT/GPT2 have a unique Swiss-Roll shaped manifold, that is not observed in other models, we look at how the manifold is related to the positional encoding in GPT2. Note that we truncate the whole PTB text into 512-length segments, and feed those segments into the models. The positional encoding is applied to each 512-length segment. We pick a few punctuation and words, and draw them in the space labeled by their relative positions in their corresponding segments. The position ID ranges from 0 to 511.

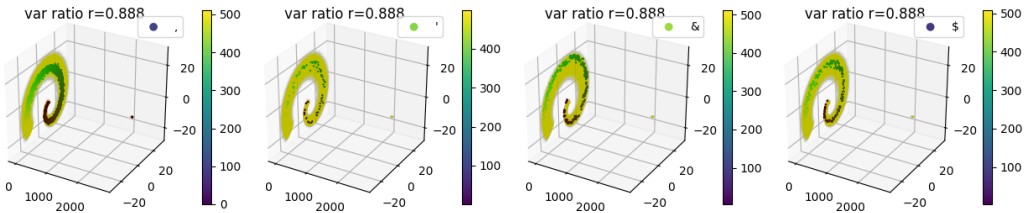

Figure 29: GPT2 Layer 3 Punctuation. The position ID is monotonically increasing along the manifold.

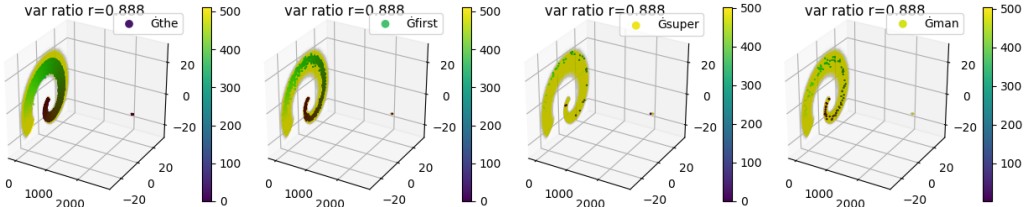

Figure 30: GPT2 Layer 3 Words. The position ID is monotonically increasing along the manifold.

We select 4 punctuation, ", ' & $", and four words "the first super man", draw them in Figure 29 30. The color bar on the side indicates the relative position ID in their segment. Darker color is smaller IDs and lighter color is bigger IDs. Clearly, for both punctuation and words, the center of the Swiss-Roll corresponds to lower position IDs, where the other end of the manifold are high IDs. Also, the distribution is monotonic. From the center to the far end, the position ID increases. This suggests that the positional encoding is indeed highly correlated with the Swiss-Roll manifold for GPT models. The reason causing this is deferred to future study.

Note that this finding is consistent with that reported in (Reif et al., 2019), where they found that positions of the token matters (tokens take all neighbors' information indiscriminately, rather than only attending up to their semantic boundaries) in the BERT embedding geometry. We also study the context/semantic influence of the embeddings in the next subsection.

### D.7 Context in the Geometry

We also look at how the context information influences the geometry. It is more sophisticated to analyze the context, so we pick a few examples to look at their context and corresponding positions in the embedding spaces. In particular, we choose the common polysemous words "like" and "interest", as two examples. The word "like" often has two different use cases: 1. favor; 2. similar to. There are also some fixed phrases such as "would like". The word "interest" has two senses as well: 1. like to do something; 2. the money sense.

We identified the target word token ("like" or "interest"), and then print out 5 tokens before and after the target, as the context for illustration in Figure 31. From the figure we are not able to identify a clear pattern that word sense is correlated with the geometric space. However, this is only inspected by manually checking a few samples. A full statistical analysis should be carried out in the future work.

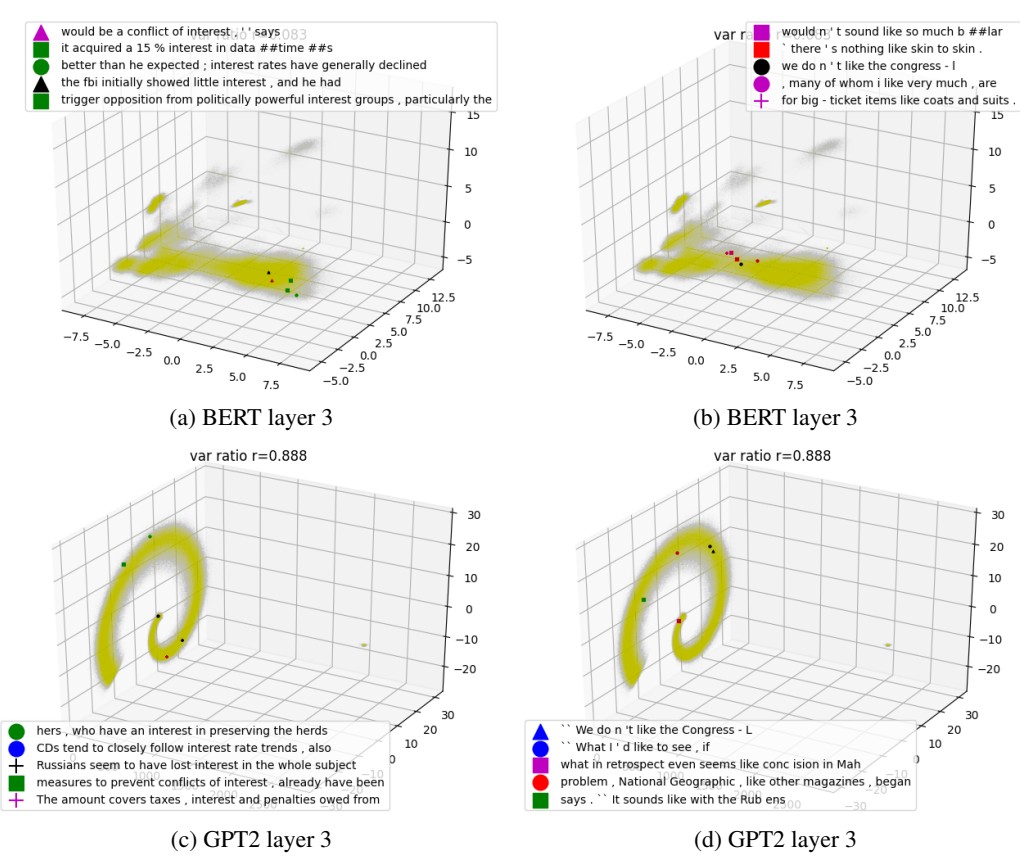

(a) BERT layer 3            (b) BERT layer 3

(c) GPT2 layer 3            (d) GPT2 layer 3

Figure 31: The context and positions in the embedding space.

