# OpenReview forum: "Isotropy in the Contextual Embedding Space: Clusters and Manifolds"
_ICLR.cc/2021/Conference — ICLR 2021 Poster_

### Official Review · AnonReviewer1 · 2020-10-27
**An exploratory analysis of contextualized embedding geometry**

**Rating:** 7
**Confidence:** 4

**Review:**

This paper analyzes the geometry of several contextualized embeddings. The authors show that global anisotropy is caused by strong clustering of word vectors, and that vectors of different word types are isotropically distributed within the cluster.

**Strengths**
- This work is a nice-to-have extension of [Ethayarajh (2019)](https://www.aclweb.org/anthology/D19-1006.pdf) that dives deeper into the geometric properties of contextualized vectors.
- The research question is clearly stated (Why doesn't anisotropy hurt performance?) and clearly answered (There's no anisotropy locally).
- The 3D visualizations provide a better geometric intuition than the flat visualizations that are common in this kind of papers.

**Issues**
- I don't think that good performance _contradicts_ anisotropy. For example, we already know that the classical static embeddings are also anisotropic [(Mimno and Thompson, 2017)](https://www.aclweb.org/anthology/D17-1308.pdf), and this means that good performance (as measured by downstream tasks) may co-exist with anisotropy. So please consider rewording the beginning of Section 1.2. For example, instead of "There is an apparent contradiction." consider "It is not clear why ..."
- How _representative_ is one random sample from $\Phi(t_i)$ for measuring $S_\text{inter}$ in formula (1). You gave an example in the Introduction when the same word type (bank) can have totally different meanings depending on context, and thus (I believe) the corresponding $\phi_1(\text{bank})$ and $\phi_2(\text{bank})$ may be totally different. Why not taking more samples for polysemous words?
- Why do you use different distance metrics (Euclidean vs cosine) for estimating LIDs of contextualized vs static embeddings (Table 3)?
- "For GPT2, we had hoped to find that some types are associated with one cluster and other types are associated with the other cluster, but that is not verified in our experiments" -- I think you should look at contexts rather than types (since you're dealing with the contextualized embeddings). It would be interesting to see whether you have the same type in both clusters, and then to look at its contexts. I bet that the contexts will differ.

**Minor issues**
- "We find a low-dimensional manifold in GPT/GPT2 embeddings, but not in BERT/DistilBERT embeddings." -- but your LIDs are low for BERT/D-BERT layers as well! Why can't you claim the low-dimensionality for BERT/D-BERT embeddigs?
- I doubt that PTB with 10K vocabulary size gives "good" coverage in 2020. You may simply state that this a widely-used dataset.
- Wiki2 (Merity et al., 2016) is usually referred to as _WikiText-2_.
- Please consider rephrasing "experiments" -> "analysis", as you are not conducting _controlled experiments_, but rather performing exploratory analysis of the embeddings.

---

> ### Author Response · Authors · 2020-11-13
> **We thank for the positive comments and very constructive suggestions.**
>
> We thank for the positive comments and very inspiring suggestions. We carefully address the issues here, and significantly update our paper by adding a new Appendix D (in the end, page 17-20) to address comments and include additional studies. After rebuttal we will merge some of them into the main paper.
>
> * Good performance does not contradicts anisotropy.
>
> A: Yes we agree, we have rephrased in the paper. Thanks for mentioning Mimno&Thompson2017, a very nice reference added to the updated version. They identified anisotropy in SGNS and found this is more related to negative sampling objective than semantics.  We have similar hypothesis (model structures/learning affects geometry, Appendix D.2, D.3) not yet fully verified. Though intuitively we hope for isotropic spaces, and Mu&Viswanath2018 proposed methods to improve performance by making it more isotropic. Still, this is not a clear conclusion so we rephrase the sentences as suggested in the updated version.
>
> * More samples for polysemous words when computing inter-cosine.
>
> A: This is a good point. Though selecting tokens based on word sense requires more work, e.g. bringing in a word sense predictor, it is still a promising direction. We plan to first get more random samples, regardless of context, and see if there is any change. The second step is to sample based on easy-to-implement rules. We are doing this and hope to update soon.
>
> * Use both Euclidean and cosine for LID.
>
> A: Simply for completeness. Euclidean distance is more used in LID and easy to think. Cosine is more popular in NLP/word-embedding domain, and reported in Aumuller&Ceccarello2019 so we can directly compare. Both gives almost identical estimates in our tests.
>
> * Look at context.
>
> A: This is a very inspiring idea. Combined with Reviewer 2's suggestion to check the positions in context, we feel this is worth investigating. Still, context-based study requires additional tools for semantic understanding, which could be a heavy lift. We will start from some manual check.
>
> * Low-dimensional manifold in BERT.
>
> A: We incorrectly states this, thanks for pointing out. Actually we mean a Swiss-Roll shaped manifold is only found in GPT but not BERT. Indeed both models have low LID indicating data are not occupying entire spaces. We correct this in the updated version.
>
> * Other minors/typos/mistakes in writing.
>
> A: We appreciate for the comments regarding writings / more precise statements. We have taken care of them accordingly in the updated version.
>
> Thank again for the constructive suggestions and we value these ideas to improve our paper.
>
> Best,
>
> Authors

---

> > ### Comment · AnonReviewer1 · 2020-11-19
> > **So far I am satisfied with the response and the paper update**
> >
> > Thank you for the quick response!
> >
> > I am satisfied with your clarifications and updates so far. Looking forward to the analysis of contexts ...

---

> > > ### Author Response · Authors · 2020-11-23
> > > **Thank you. New update on position and context analysis.**
> > >
> > > Thank you for your positive feedback.
> > >
> > > We have an update that briefly analyze positional encoding and context effects on the geometry. They are in Appendix D.6 and D.7 (page 21-22, in the end of the paper).
> > >
> > > In short summary, after manually checking a few examples, we are not able to identify clear pattern that relates context to the embedding shape (2 polysemous words, 'like' and 'interest' with context, are illustrated in page 22). One reason could be that we didn't get enough resolution for a single token's context, as we only hand pick examples due to time limit.
> > >
> > > However, we find that, at least for GPT2, the positional encoding is strongly correlated with the Swiss-Roll manifold shape. Position ID ranges from 0 to 511. The lower ID value, the closer to the center of the roll. The position ID is monotonically increasing as going from center to edge along the manifold (page 21).
> > >
> > > Future work would include more statistical analysis on context and positions as well. Thank again for the interesting ideas and valuable thoughts. We appreciate for your precious time reviewing this work.
> > >
> > > Best,
> > >
> > > Authors.

---

> > > > ### Comment · AnonReviewer1 · 2020-11-23
> > > > **Thank you for the update**
> > > >
> > > > Thank you for inspecting the contexts. I'd like to keep my score at 7. I am sure that the paper should be accepted, but I am not sure if it should be in the top-50% of accepted papers.

---

> > > > > ### Author Response · Authors · 2020-11-24
> > > > > **Sure, we appreciate that.**
> > > > >
> > > > > Thanks for the quick reply. We understand and appreciate that. It is our pleasure to present the findings.
> > > > >
> > > > > Best,
> > > > >
> > > > > Authors

---

> > > > ### Comment · AnonReviewer4 · 2020-11-23
> > > > **Swiss Roll manifold**
> > > >
> > > > If positional encoding is related to the Swiss Roll, then a 2d projection of the manifold could look a lot like the spiral manifold noticed by https://arxiv.org/abs/1906.02715 (Andy Coenen et al.). Similarly, this might be a direct effect of the particular positional encoding used. May be worth making this connection explicitly.

---

> > > > > ### Author Response · Authors · 2020-11-24
> > > > > **That's true. Updated.**
> > > > >
> > > > > We agree.
> > > > >
> > > > > The findings of the positional encoding effects, is consistent with the low-dimension manifold suggested in that paper. They also noticed that tokens tends to ignore semantic boundaries but absorb all info from neighbors.
> > > > >
> > > > > We have cited that paper (the NeurIPS version, reif2019visualizing) in related work for that they identified the semantic trees. Now we also connect this positional-encoding manifold part with that paper in our updated version.
> > > > >
> > > > > Thanks for pointing out this. Different positional encoding (pre-defined sine or fully learned) and how they affect geometry is certainly worth further study now.
> > > > >
> > > > > Best,
> > > > >
> > > > > Authors

---

### Official Review · AnonReviewer4 · 2020-10-27
**A paper of limited novelty, which could be improved by deeper analysis.**

**Rating:** 7
**Confidence:** 3

**Review:**

Findings:
- Reproduces various existing findings about anisotropy of contextual representations viewed globally.
- Contextual representations are highly isotropic within clusters of the representations.
- GPT representations follow a Swiss Roll manifold, where the most frequent words appear at the head and less frequent words are gradually appended at the bottom.
- The Swiss Roll manifold is taller in deeper layers.
- BERT representations fall in a Euclidean space.
-  The Local Intrinsic Dimension of contextual embeddings is lower than for unigram embeddings.

Pros:
- The manifold analysis of word frequency is intriguing and intuitive.
- The explanation of experiments was clear in each section.
- They produce compelling evidence that the global token-level anisotropy of these representations is largely due to membership of large clusters. This is a valuable contribution because it explains previous findings in Ethayarajh 2019 and reconciles them with theoretical expectations.

Cons:
- In Section 2.3 and Section 3.1, the paper gives insufficient credit to Ethayarajh 2019. As far as I could tell, every initial result is a reproduction of a result from Ethayarajh 2019, and the methods are very similar.
- Though they acknowledge it, the methodology is largely taken from existing unigram embedding analysis.
- Once they start to identify very well defined clusters, I was very curious about the distinctions between the islands. It would not be difficult to inspect some of the data by hand, so I don't understand why the authors didn't try.
- The authors offer no analysis for the difference in behavior between different models. I felt like I was reading a taxonomy, and the plots were left for the reader to connect. The authors have presumably spent quite a while thinking about these geometric structures and models, so surely they have conjectures about the behavior they observed or hypotheses they can test.

Minor / style:
- I didn't realize until the conclusion that your main finding was an explanation of existing claims about anisotropy by considering behavior within the clusters, so that needs more emphasis.
- The papers you cite do a decent job of explaining why isotropy in the representation space is significant both token- and type-wise. The paper would be a lot more readable if you made a similar effort in explaining background.
- There is inconsistent use of "isotropy" vs "isotropicity".
- Citations needed for claim "widely believed that the contextual space is so weird"
- Needs proofreading for minor grammar and typos (e.g., downstreaming instead of downstream, could resides instead of could reside)
- Instead of referring variously to high similarity or cosine, silhouette scores, and other measurements of isotropy, it might be clearer to link each concept to isotropy once, and then each subsequent result simply refer to it as isotropy while mentioning the metric. Then the reader doesn't have to constantly remember which metric indicates high isotropy as they read the results.

Questions:
- The authors claim to select the clusters that maximize MMS. I read the wording to imply that this optimum is tractable/stable. Is that the case?

(Number rating was updated from 3 to 7 in light of substantial expanded experiments and analysis.)

---

> ### Author Response · Authors · 2020-11-13
> **We thank for the valuable criticisms, the constructive suggestions, and carefully clarify/address the raised issues. We significantly updated our paper.**
>
> We thank for the valuable criticisms, the constructive suggestions, and carefully clarify/address the raised issues. We significantly updated our paper by adding a new Appendix D (in the end, page 17-20) to address comments and include additional studies. After rebuttal we will merge some of them into the main paper.
>
> * Question on tractable/stable MMS estimate.
>
> A: We answer in two folds:
>
> (1) MMS is tractable and stable in our setting. It is tractable by restricting search range to [2,15], and use 20k random samples to estimate the score. We report average over 5 runs and the std are small (~ 1e-3, for strong separation like GPT2, std is less than 1e-4), showing 20k sample size is stable. Small clusters could be missing due to sampling, but large ones will be kept.
>
> (2) K-Means is indeed problematic, but OK. (a) K-Means is sensitive to initialization, but we are not seeking for optimal clustering. Sub-optimal, e.g. treating two overlapping clusters as a big one, is totally fine. To illustrate this, we add another metric to select slightly different Ks, and yield very similar cosines in the updated Appendix D.1 (page 17). (b) K-Means implicitly assumes convex clusters, which often does not hold. However, density-based clustering is too slow for these datasets, so it is a trade-off and still useful to distinct highly-seperated clusters.
>
> Details on this discussion is added to the new Appendix D.1.
>
> * Insufficient credit to Ethayarajh2019. Initial result is a reproduction.
>
> A: Must be a misunderstanding, we give high credit to Ethayarajh2019. We apologize for any unclear statements. Ethayarajh2019 motivates our paper, and we refer it many times throughout the paper. Also, as explicitly written in the paper, we reproduce and validate Ethayarajh2019's results in Section 2.3. This is the motivation and leads to our following analysis. We make this more clear in the updated version.
>
> * Methodology is largely taken from existing analysis. Limited novelty.
>
> A: Our clustering analysis part is inspired by Mu&Viswanath2018 and Ethayarajh2019. We acknowledge that and point out the key differences in Section 1.2. Additionally, the manifold part is new and not based on these works. Please consider this paper not as a new methodology paper, but an exploratory analysis.
>
> * Analysis on distinctions between islands.
>
> A: We tried this. We hand-picked tokens, and try to see how they are distributed. Taking BERT as an example, high-frequent and low-frequent words take opposite sides of the main island. Punctuations are random and less concentrated, but they occupy those islands. For instance, '!' and '`' are distinct islands, while some others are on the main cluster. For GPT2, almost all single letters (a to z), and mid-to-high frequent words, occupy both the left (big) and right (small) islands. We didn't find any token that only appears in the right small island. It seems that the tokens in the small island always have mirrors in the left big cluster. We add these discussions and examples in Appendix D.2 (page 17).
>
> So we only have a conclusion that word distributions in the space, are correlated with the word frequency (also reported in the paper Section 4.3 as well). We are not clear yet what causes this. We do have a hypothesis that high frequent tokens are updated many times and the learning process can better align them to be close to isotropic. This needs future work to validate. We also don't have a clear semantic conclusion for the clusters now, which needs more statistical analysis. Hope Appendix D.2 could provide a more clear illustration.
>
> * Analysis on different models' behaviors and relationship to their geometric structures.
>
> A: Indeed we did this. We tried to establish a systematic view of different types of models. In fact, we also studied XLM model, which is a multi-lingual translation model. We found it does not have clear-separated clusters and the embedding is isotropic and sit at the origin. We have hypothesis that cross-lingual model are trained to force the embedding to be shared by different languages, thus a centered isotropic embedding shape is formed during training. We add this part in updated version, Appendix D.3 (page 19). But since we only looked at one typical model in each type (masked LM, causal LM and translation LM), we are not confident to draw conclusions. This requires more study on a large range of models, until then we only have hypothesis. We add the XLM study in Appendix. We also try to use the shared embedding to explain that high frequent words' distributions are less diverged. This needs future validations as well.
>
> * Minor/Style:
>
> A: We appreciate all these suggestions and have already taken care of most of them in the updated version.
>
> Thanks again and please consider our clarifications / discussions above and the updated experiments (Appendix D). Hope these could address your concerns and let you reconsider the rating of this paper.
>
> Best,
>
> Authors

---

> > ### Comment · AnonReviewer4 · 2020-11-13
> > **Thank you, very satisfied with your additions!**
> >
> > Regarding the "exploratory" vs. "methodological" question: I mentioned the novelty of the methodology because I was disappointed with the exploration. Many of the results were replications using similar methods, and in the second half when new results started to emerge, (what I saw as) obvious leads were not followed up. Therefore, the exploration was incomplete and I did not feel it should be published as-is, which was the main reason for such a low score. With your additions, you have very quickly addressed the glaring gaps in analysis, so thank you for that. I understand that confirmation of any hypotheses about model comparison are outside your scope, but I'm happy to see your thoughts on it in a future work or discussion section.
> >
> > If I were to re-score this paper now I'd give it a much higher score.

---

> > > ### Author Response · Authors · 2020-11-14
> > > **Thanks a lot for the positive feedback on the updated version!**
> > >
> > > We are very happy to see the positive feedback on our updated paper. We highly appreciate for all those comments and suggestions to improve our paper. Thanks for your understanding on the model-type comparisons as well.
> > >
> > > Due to time limit, we will try our best to continue conducting a few more experiments, and merge some of the Appendix D into the main paper. It is our pleasure to present this work, and we appreciate for your precious time to review this work.
> > >
> > > Best,
> > >
> > > Authors

---

### Official Review · AnonReviewer2 · 2020-10-28
**Thorough, thought-provoking, a bit unfocused and inconclusive**

**Rating:** 7
**Confidence:** 4

**Review:**

#### Summary:

The authors investigate the token embedding space of a variety of contextual embedding models for natural language. Using techniques based on nearest neighbors, clustering, and PCA, they report a variety of results on local dimensionality / anisotropy / clustering / manifold structure in these embedding models which are of general interest to scientists and practitioners hoping to understand these models. These include findings of (local) isotropy in the embeddings when appropriately clustered and shifted, and an apparent manifold structure in the GPT models.

#### Reasons for score:

This is a generally thorough and well-executed paper, and a careful examination of these embedding models is of great general interest to the ICLR community.
However, I feel the analysis is a bit shallow at certain points and consists of reporting (interesting) findings without necessarily delving deeper or adequately explaining them.  I think this paper presents a great jumping off point for further research on the subject, as it certainly raised quite a few questions with me. I support its acceptance but would hope to see the authors address some of the questions raised here.

#### Positives:

- Lots of analysis with several different techniques

- Very interesting and relevant subject area

- Thorough use of recent related work

- The paper is quite well written and organized considering the inherent challenge of writing up research of this sprawling nature.

#### Concerns / Comments:

- Noting the very different behavior of GPT from the other representations — could this be due to the learned positional encodings? This might also explain the Swiss-roll style paths seen when examining the different embeddings of the same word types. Could position along those curved paths be correlated to sentence position of the tokens?

- In this spirit, it would be great to get a better characterization of what the different clusters correspond to, especially in the case of GPT. Could there be a better investigation of the relationship between predictive / causal and non-causal models and these clusterings?

- More generally, the “Swiss roll” observation is intriguing, but since it only appears in one family of models that has a very similar (transformer) architecture to models in which it does not appear, what are we to make of it?

- How can the intrinsic dimensionality at each layer increase with depth? Considering each layer as living on a manifold, the transformation at each layer should act as a coordinate chart for the next layer’s manifold, which should only allow a reduction in dimension. Unless I am missing something, that suggests that these estimators do not have enough samples and/or are measuring something different than the dimension of a manifold. Section 4.4 should explain how this could be happening.

- More generally, I find the low LIDs in GPT hard to understand or interpret without more analysis, and would also like to understand the very large first dimension of the GPT models.

- It would be great to see more suggestions / takeaways for practitioners. What, if anything, does local (an)isotropy imply for deep learning researchers doing work in this area?

---

> ### Author Response · Authors · 2020-11-13
> **We thank for the positive comments and very inspiring suggestions.**
>
> We thank for the positive comments and very inspiring suggestions. We carefully address the issues here, and significantly update our paper by adding a new Appendix D (in the end, page 17-20) to address comments and include additional studies. After rebuttal we will merge some of them into the main paper.
>
> * GPT context space is very different, might due to positions.
>
> A: Very inspiring idea. We investigated another thing: the cosines vs positional embedding, and found the positional embedding lead to a periodic cosines (peak at every 512 offset, as we use 512 length input). But we didn't connect the positional embedding with the manifold curves. We are conducting this study, combined with Reviewer 1's suggestion to look into the context. We will update soon.
>
> * Does causal/non-causal model have this behavior.
>
> A: This was indeed our hypothesis. Unfortunately we didn't find enough evidence. In fact, we try to build a systematic view, by looking at 3 types of models: masked LM (MLM, e.g. BERT), causal LM (CLM, e.g. GPT), translation LM (TLM, e.g. XLM). We have XLM studies in Appendix D.3 now, which shows an isotropic and concentrated embedding space. We suspect it is due to shared embedding across languages in XLM. Nevertheless, since too few models are inspected for each type, we are not confident to draw the conclusion on model structures with geometric properties.
>
> * What to do with Swiss Roll as it only appears in one model family.
>
> A: This is not yet clear to us. We hope to investigate more casual models to establish the connection between casualty and this special shape. The ELMo is causal but not transformer based, thus no such pattern identified. We believe this Swiss Roll is too special to be just coincidence. Once identified the reason (casualty, training process, tokenization, semantic relations?), it should be insightful for new model designs.
>
> * Local intrinsic dimension increases with depth, due to not enough samples.
>
> A: (1) Indeed we checked how different number of samples affects the approximation. The estimate is very consistent when we range from 100 to 1000 samples, so we just use 100 to be consistent with Aumuller&Ceccarello2019. We add these detailed experiments in Appendix D.4 (page 20) in the updated paper.
>
> (2) If 2nd layer is only a subspace of 1st layer, and so on, then dimension indeed can only reduce. However, note that all layers stay in the original 768-D space, and transformation by the network won't force the next layer's points to stay within previous subspace. In other words, it can expand, at a cost of losing density. For example, the spiral band (1-D) in GPT's front layer, becomes a Swiss Roll (2-D), and the roll surface get thickness (3-D), as layer goes deeper. The increasing LID is validated using different number of samples, in Appendix D.4 now. But we are not clear about the reasons yet, only suspect that data is spreading as more context info is added in later layers (embedding for token in deeper layer is based on summation of all embeddings in the context, due to attention). We add this discussion to the updated version.
>
> * The 1st principle dimension in GPT.
>
> A: In GPT2, the 1st principle dimension makes a clear separation of the two clusters in the space. As layer goes deeper until the very last layer, the distance between the two islands increases. We are not yet clear what causes this. More analysis on the tokens and several observations are now in Appendix D.2 (page 17). Most mid-to-high frequent word are appearing on both islands, and no tokens only appear on the small cluster. We are doing more analysis based on your suggestion to check positions in context, and hope to find some missing connections.
>
> * Suggestions and takeaways.
>
> A: One promising direction (we are currently experimenting) is to study if turning local isotropy into global, will benefit model training and performance. In other words, content regularization could be added, as done in Mu&Viswanath2018 for static space. We add this discussion in the updated version as we have one more page now.
>
> Thank again for the constructive suggestions.
>
> Best,
>
> Authors

---

> ### Author Response · Authors · 2020-11-23
> **Update: new version with positional encoding.**
>
> We have updated the paper and add Appendix D.6 and D.7 to study the positional encoding's and context's influence on the space.
>
> We pick a few samples and find that, at least for GPT2, the positional encoding is strongly correlated with the Swiss-Roll manifold shape. Position ID ranges from 0 to 511. The lower ID value, the closer to the center of the roll. The position ID is monotonically increasing as going from center to edge along the manifold (page 21).
>
> This is an interesting finding and we will study the reason behind this in future work. Thank again for the inspiring ideas. We appreciate for your precious time reviewing this work.
>
> Best,
>
> Authors.

---

### Decision · Program_Chairs · 2021-01-07
**Final Decision**

**Decision:**

Accept (Poster)

**Comment:**

This paper presents a broad exploratory analysis of the geometry of token representations in large language models, with a focus on isotropy and manifold structure, and reveals some surprising findings that help explain past observations.

Pros:
- Clear and surprising analytical findings concerning a broad and widely-used family of models.

Cons:
- The paper is a fairly broad exploratory analysis, with no single precise claim that ties together every piece of the work.

I thank both the authors and reviewers for an unusually productive discussion.